# A meta-analysis of the association between male dimorphism and fitness outcomes in humans

Linda H Lidborg[1]\*, Catharine Penelope Cross[2], Lynda G Boothroyd[1]

[1]Department of Psychology, Durham University, Durham, United Kingdom; [2]School of Psychology and Neuroscience, University of St Andrews, St Andrews, United Kingdom

**Abstract** Humans are sexually dimorphic: men and women differ in body build and composition, craniofacial structure, and voice pitch, likely mediated in part by developmental testosterone. Sexual selection hypotheses posit that, ancestrally, more 'masculine' men may have acquired more mates and/or sired more viable offspring. Thus far, however, evidence for either association is unclear. Here, we meta-analyze the relationships between six masculine traits and mating/reproductive outcomes (96 studies, 474 effects, N = 177,044). Voice pitch, height, and testosterone all predicted mating; however, strength/muscularity was the strongest and only consistent predictor of both mating and reproduction. Facial masculinity and digit ratios did not significantly predict either. There was no clear evidence for any effects of masculinity on offspring viability. Our findings support arguments that strength/muscularity may be sexually selected in humans, but cast doubt regarding selection for other forms of masculinity and highlight the need to increase tests of evolutionary hypotheses outside of industrialized populations.

\*For correspondence:
lhlidborg@gmail.com

**Competing interest:** The authors declare that no competing interests exist.

## Editor's evaluation

This paper presents a series of meta-analyses to test the plausibility of sexual selection hypotheses for the origins and/or maintenance of six sexually differentiated traits in humans, with strength/muscularity found to be significantly associated with both mating and reproduction. The authors considered both published and unpublished datasets to help account for potential publication biases that could theoretically impact meta-analysis results, which is a particular strength of this study.

## Introduction

### Sexual dimorphism and masculinity in humans

Sexual dimorphism refers to sex differences in morphological and behavioral traits, excluding reproductive organs (*Plavcan, 2001*), with particular emphasis on traits thought to have evolved through sexual selection (*Crook, 1972*). Humans are a sexually dimorphic species (*Plavcan, 2001*). Sexual selection in mammalian species, including human and non-human primates, is commonly argued to have acted more strongly on male traits, as a consequence of greater variance in males' reproductive output (*Hammer et al., 2008*) and a male-biased operational sex ratio, that is a surplus of reproductively available males relative to fertile females (e.g. *Mitani et al., 1996*).

Dimorphic traits that are exaggerated in males are typically referred to as masculine. In humans, masculine faces are characterized by features such as a pronounced brow ridge, a longer lower face, and wider mandibles, cheekbones, and chins (*Swaddle and Reierson, 2002*). Men are, on average,

**eLife digest** Many species show sexual dimorphism: traits that are different or more exaggerated in either females or males. These traits are often thought to have evolved because they increase an individual's chances of producing offspring. While the evolution of male dimorphism – often referred to as masculinity – is generally well understood in many animal species, opinions differ as to whether such traits also increase male reproduction in humans.

Lidborg et al. tried to shed light on the evolution of masculine traits in human males (such as a more robust-looking facial structure, and increased strength and muscularity) by testing whether men with these traits reported having more sexual partners and/or whether they had more children compared to men in which these traits were not as extreme. To do so, Lidborg et al. compiled previously published data from populations all across the world and tested the associations between the traits and both partner numbers and reproduction.

The results showed that men who were physically stronger and more muscular reported having more sexual partners and, in societies that do not use contraception, these men also had more children than other men. Lidborg et al. also found that in industrialized societies, men who were taller, had a lower voice pitch and higher testosterone levels also reported more sexual partners, but they did not produce more offspring. Lastly, the analysis showed that men with more robust facial structures faces did not report having more partners or more children.

These findings suggest that traits such as strength and muscle mass in men may be favoured by evolution. Importantly, this seems to be the case across all societies from which Lidborg et al. analyzed data. The results also show that some of the traits Lidborg et al. tested – such as being tall – might increase the number of partners men in industrialized countries have, but not the number of children men in more traditional societies (such as hunter-gatherers) produce. This could be because women's preferences for men's traits differ between cultures.

Ultimately, Lidborg et al.'s analysis suggests that across different cultural contexts, only strength and muscularity truly do seem to matter for men's mating and reproduction.

7–8% taller than women (*Gray and Wolfe, 1980*) and weigh approximately 15% more (*Smith and Jungers, 1997*). Relative to this fairly modest body size dimorphism, upper body musculature and strength are highly dimorphic in humans: compared to women, men have 61% more overall muscle mass, and 90% greater upper body strength (*Lassek and Gaulin, 2009*). Men's bodies also tend to have a V- or wedge-shape, showing a greater shoulder-to-hip ratio (*Hughes and Gallup, 2003*; *Singh, 1993*) and waist-to-chest ratio (*Maisey et al., 1999*; *Weeden and Sabini, 2007*) than women's. Second-to-fourth finger (digit) length ratios are often claimed to be sexually dimorphic, with men's 2D:4D typically being lower than women's (*Manning, 2002*, although this may not be universal: *Apicella et al., 2016*). In addition, fundamental frequency, commonly referred to as voice pitch, is nearly six standard deviations lower in men than in women (*Puts et al., 2012*).

The development of these masculine traits in men is influenced by exposure to androgens, particularly testosterone. With the exception of 2D:4D, which is commonly claimed to be influenced primarily by prenatal testosterone levels and is present at birth (*Galis et al., 2010*; but see *Richards et al., 2019*), masculine traits generally develop or become exaggerated following a surge in testosterone production at sexual maturity (*Butterfield et al., 2009*; *Fechner, 2003*; *Weston et al., 2007*) – although it is not necessarily clear whether the size of that surge corresponds directly to the extent of trait expression.

## Proposed mechanisms underlying the evolution of masculine traits

Key to the assumption that men's masculine traits are sexually selected is that masculine traits should be reliably associated with greater biological fitness. Men may increase fitness by producing a greater quantity of offspring overall (i.e. greater *fertility*), by acquiring a greater number of partners which may in turn mediate offspring numbers (greater *mating success*), and/or by producing more *surviving* offspring (greater *reproductive success*).

Two key hypotheses and attendant mechanisms have been drawn on by evolutionary behavioral scientists, predicting positive associations between masculinity and fitness outcomes. Firstly, according

to the *immunocompetence handicap hypothesis* (*Folstad and Karter, 1992*), masculine traits are a costly signal of heritable immunocompetence, that is good genetic quality, due to the putative immunosuppressive properties of testosterone (see *Muehlenbein and Bribiescas, 2005*). Masculine men should therefore produce healthier and more viable offspring, who are more likely to survive. Thus, women should be able to increase their fitness (via offspring survival) by selecting masculine men as mates. Authors therefore suggested that masculinity in men is intersexually selected, evolved and/or maintained through female choice, and should be associated with greater mating success in contexts where women are able to exercise choice. This should thus result in greater reproductive success, and an advantage in offspring survival.

The immunocompetence handicap hypothesis has persisted in the literature, particularly with reference to facial masculinity (although there are no a priori reasons to expect this putative mechanism to act more strongly on men's faces than on their bodies), despite concerns regarding its validity since at least 2005 (*Boothroyd et al., 2005*). While beyond the scope of this article, common criticisms include that the relationship between testosterone and health is complex (*Nowak et al., 2018*), and facial masculinity is inconsistently linked to health (e.g. *Boothroyd et al., 2013*; *Foo et al., 2020*; *Marcinkowska et al., 2019*; *Scott et al., 2013*; *Zaidi et al., 2019*). Evidence is similarly mixed regarding the key assumption that women are attracted to masculinity in men's faces (*Boothroyd et al., 2013*; *Zeigler-Hill et al., 2015*) and bodies (*Frederick and Haselton, 2007*; *Gray and Frederick, 2012*; *Lukaszewski et al., 2014*; *Sell et al., 2017*).

Secondly, under the *male-male competition hypothesis,* authors have argued that formidable (i.e. physically strong and imposing) men are better equipped to compete with other men for resources, status, and partners (*Hill et al., 2016*; *Puts, 2016*), through e.g. direct physical contests or by deterring rivals indirectly (*Hill et al., 2016*; *Sell et al., 2012*). For instance, increased musculature may intimidate competitors by signaling fighting prowess (*Sell et al., 2009*) and strength (*Durkee et al., 2018*), while facial masculinity and voice pitch may also have an indirect relationship with perceived formidability (*Butovskaya et al., 2018*; *Haselhuhn et al., 2015*; *Raine et al., 2018*; *Little et al., 2015*; *Puts and Aung, 2019*; *Scott et al., 2014*). Importantly, while male-male competition is often framed as an alternative to female choice, women may preferentially mate with both well-resourced men, and with competitive men, facilitating intersexual selection for masculinity (i.e. a 'sexy sons' effect, see *Weatherhead and Robertson, 1979*) where male status is due to, or competitiveness is cued by, formidability (*Scott et al., 2013*). Some authors have suggested that formidability increases men's mating success through dominance over other men (which may create the circumstances that women select them as mates) rather than women's direct preferences for formidable traits per se (*Hill et al., 2013*; *Kordsmeyer et al., 2018*; *Slatcher et al., 2011*). However, regardless of whether the driving mechanism is intra- or intersexual selection (or a combination thereof), the male-male competition hypothesis predicts that formidable men will acquire more partners over their lifetime, which will in turn result in more offspring. This approach, however, does not make any particular predictions regarding offspring health or survival.

It can be noted that proponents of both the immunocompetence and male-male competition hypotheses have also suggested that more masculine men may show reduced investment in romantic relationships and in offspring (*Booth and Dabbs, 1993*; *Boothroyd et al., 2007*; *Muller et al., 2008*; *Schild et al., 2020*), potentially suppressing offspring health/survival. This could arise from an association between circulating testosterone (which masculine traits are commonly argued to index) and motivation for sexual behavior (*Grebe et al., 2019*; *Halpern et al., 1993*) shifting effort away from parental investment toward pursuit of mating opportunities. Two important caveats here, however, are that the relationship between men's testosterone levels in adolescence (when most masculine traits become exaggerated) and in adulthood is exceedingly weak (*van Bokhoven et al., 2006*), and masculine trait expression in adulthood is not consistently correlated with adult testosterone levels (e.g. *Lefevre et al., 2013*; *Peters et al., 2008*). Simply being more attractive to potential new partners, however, might shift behavior away from relationship investment (for discussion see e.g. *Gangestad and Simpson, 2000*). Because of this, many authors have previously suggested that women face a trade-off between the (health or competitive) benefits of masculinity, and paternal investment.

## The association between masculine traits and biological fitness

We therefore have at least two theoretical positions which assert that masculine men should have greater numbers of sexual partners, greater offspring numbers, and perhaps a greater proportion of surviving offspring, in at least some circumstances. Studies addressing these predictions in societies without effective contraception have done so directly via offspring numbers and/or offspring survival. In most industrialized populations, where access to contraceptives attenuates the relationship between sexual behavior and reproductive success, mating success measures are often used instead. These include preferences for casual sex, number of sexual partners, and age at first sexual intercourse (earlier sexual activity allows for a greater lifetime number of sexual partners), as these are assumed to have correlated with reproductive success in men under ancestral conditions (*Pérusse, 2010*).

A key problem, however, is that the predictions outlined above do not always capture the diversity of human reproductive ecologies even where diverse data exists. We have already noted the fact that female choice may be important to outcomes above. Furthermore, even amongst non-contracepting populations, differences in rates of polygyny, pair-bond breakdown, and attitudes to fertility may moderate reproductive success and its variance. For instance, monogamous cultures do not typically show greater variance in men's versus women's reproductive success (*Brown et al., 2009*) and while increasing numbers of sexual partners (e.g. in serially monogamous or polygynous cultures) may often be important for increasing male reproductive success, the inverse is true amongst the Pimbwe where women are more advantaged by increased numbers of partners (*Borgerhoff Mulder, 2009*). Similarly, although the strongly monogamous Agta show high rates of fertility (*Boothroyd et al., 2017*), data from ostensibly non-contracepting rural Catholics in C20th Poland (*Pawlowski et al., 2008*) shows much lower rates of fertility. These issues highlight the fact that humans have likely had diverse reproductive and pair-bonding norms for a long time. As such we can make two observations. Firstly, availability of contraception in low-fertility samples might 'free' sexual behavior from the constraints of pregnancy avoidance, and we might find *stronger* relationships between any evolved motivation for sex, and actual sexual behavior, in these samples than would have necessarily been found ancestrally. Secondly, however, any adaptation which has been maintained across recent hominid lineages must have been adaptive *on average* across diverse reproductive ecologies. As such, if the proposed adaptation (masculinity leading to enhanced reproductive success via mating, and possibly increased offspring survival) exists, we should expect to see both: *i*. masculinity being associated with increased mating success in both high and (perhaps especially) low fertility populations, and *ii*. masculinity being on average positively associated with fertility, and potentially offspring survival, in non-contraception/high fertility populations.

## Meta-analysis in sexual selection

Meta-analysis can be a valuable tool in understanding overall patterns in evolutionarily relevant traits, both across and within species. Jennions and colleagues (*Jennions et al., 2012*) noted that many traits hypothesized to predict male mating success had not been subject to meta-analysis, and further argued that while such meta-analyses can be valuable in clarifying the nature and extent of selection for some traits, at other times they act to refute prior assumptions. They say: "A general insight from sexual selection meta-analyses … is that it is easy to be misled by a few high-profile studies into believing that a prediction is well supported. Support is often weaker than assumed." (p.1139). This point does not just apply to comparative research, but is relevant to human sexual selection work specifically. For instance, *Van Dongen and Gangestad, 2011* found that evidence for health benefits of symmetry were weaker and harder to demonstrate meta-analytically than they would have supposed, given the size of the extant literature. Similarly, when two meta-analyses into the effects of menstrual cycle on women's behavior, mate preferences, and attractiveness reached opposing conclusions (*Gildersleeve et al., 2014a*; *Wood et al., 2014*), the exercise suggested that some cycle effects were unlikely to be robust. Indeed, the more cautious analytical methods (e.g. treating unknown null results as zero rather than excluding them from analysis) resulted in a null overall effect – a finding that was later borne out by multiple large, pre-registered, studies (*Jones et al., 2018*; *Jünger et al., 2018*; *Marcinkowska et al., 2018*). The authors of the meta-analysis that found a null effect suggested that publication and inclusion bias was a particular problem in the field (*Harris et al., 2014*), although others argued against this (*Gildersleeve et al., 2014b*).

In terms of the current topic, previous studies explicitly testing the relationships between masculine traits and fitness outcomes have been overwhelmingly conducted in low fertility samples and have produced a mixture of positive, negative, and null results (e.g. *Boothroyd et al., 2017*; *Arnocky et al., 2018*; *Rhodes et al., 2005*). This creates a clear need for meta-analytic comparison of evidence from as wide a population sample as possible. To date, however, meta-analytic analyses are rare, typically exclude many aspects of masculinity, and focus on *either* mating *or* reproductive outcomes, despite both being relevant to testing the theories above. *Van Dongen and Sprengers, 2012* meta-analyzed the relationships between men's handgrip strength (HGS) and sexual behavior in only three industrialized populations (showing a weak, positive association [$r = 0.24$]). Across 33 non-industrialized societies, *von Rueden and Jaeggi, 2016* found that male status (which included, but was not limited to, measures of height and strength) weakly predicted reproductive success (overall $r = 0.19$). In contrast, Xu and colleagues (*Xu et al., 2018*) reported no significant association between men's height and offspring numbers across 16 studies when analyzing both industrialized and non-industrialized populations. Lastly, Grebe and colleagues' (*Grebe et al., 2019*) meta-analysis of 16 effects – the majority of which came from Western samples – showed that men with high levels of circulating testosterone, assayed by blood or saliva, invested more in mating effort, indexed by mating with more partners and showing greater interest in casual sex ($r = 0.22$). Across all of their analyses (which also included pair-bond status, fatherhood status, and fathering behaviors), Grebe and colleagues found no significant differences between 'Western' and 'non-Western' samples, but their 'non-Western' grouping for the relevant analysis only included a low fertility population in 21st Century China. To our knowledge, facial masculinity, voice pitch, and 2D:4D have never been meta-analyzed in relation to mating and/or reproduction.

> ## Box 1. Search terms for meta-analysis study discovery.
>
> (masculin* OR "sexual dimorphism" OR "sexually dimorphic" OR width-to-height OR muscularity OR shoulder-to-hip OR chest-to-waist OR "digit ratio" OR 2d:4d OR "hand grip strength" OR "handgrip strength" OR "grip strength" OR testosterone OR "voice pitch" OR "vocal pitch" OR voice OR "non-fat body mass" OR "lean body mass" OR "fundamental frequency" OR "facial* dominan*" OR height OR "sexual dimorphism in stature" OR "CAG repeat*") AND ("sex* partner*" OR "short-term relationship*" OR "short term mating" OR "extra pair" OR sociosexual* OR "age of first intercourse" OR "age of first sexual intercourse" OR "age at first intercourse" OR "age at first sexual intercourse" OR "age of sexual debut" OR "age at first sex" OR "mating success" OR "number of offspring" OR "offspring number" OR "number of children" OR "number of grandoffspring" OR "number of grand offspring" OR "offspring health" OR "offspring mortality" OR "mortality of offspring" OR "surviving offspring" OR "offspring survival" OR "reproductive onset" OR "reproductive success" OR "long-term relationship*" OR "age of first birth") AND (human OR man OR men OR participant*).

## The present study

The present article therefore searched widely for published and unpublished data to meta-analyze the relationships between six main masculine traits in men (facial masculinity, body masculinity, 2D:4D, voice pitch, height, and testosterone levels) and *both* mating and reproductive outcomes, in *both* high and low fertility samples. By including multiple traits, a broad search strategy, and considering high and low fertility samples both separately and together, we can ascertain whether the current scientific evidence base provides plausible support for the sexual selection of masculine traits in humans. By further testing the publication status of each effect (whether the specific effect size/analysis was reported in a published article or not), we can also evaluate the evidence for publication bias, since this is known to artificially inflate effects in diverse literatures.

Mating measures included behavioral measures such as number of sexual partners, number of marital spouses, and age at first sexual intercourse. Since increased mating effort is an additional possible route to increased reproductive output, we also included mating attitudes, such as preferences for casual sex. Reproductive measures included: fertility measures, such as number of children/grandchildren born and age at the birth of the first child; and reproductive success measures, that is number of offspring surviving childhood. Since offspring mortality is a measure specifically of offspring viability, we included this as a separate measure (i.e. mortality rate and/or number of deceased offspring).

## Materials and methods
### Literature search and study selection

A systematic search was initially carried out between November 2017 and February 2018 using the databases PsycINFO, PubMed, and Web of Science; the searches were saved and search alerts ensured inclusion of subsequently published studies. Search terms are given in *Box 1*.

Studies were also retrieved through cross-referencing, citation searches/alerts, and by asking for data on social media. The systematic search generated 2,221 results, including duplicates, and a further approximately 300 articles were found by other means. After scanning titles and abstracts, 280 articles/dissertations were reviewed in full. Studies submitted up to 1 May 2020 were accepted. Eligible studies included at least one of the following predictors: facial masculinity, body masculinity (strength, body shape, or muscle mass/non-fat body mass), 2D:4D, voice pitch, height, or testosterone levels. The following outcome measures were included:

- Mating domain: global sociosexuality (i.e. preferences for casual sex: *Penke and Asendorpf, 2008*; *Simpson and Gangestad, 1991*) and specific measures of mating attitudes and mating behaviors where:

  i.   Mating attitudes included: preferences for short-term relationships, and sociosexual attitudes and desires.
  ii.  Mating behaviors included: number of sexual partners, one-night-stands/short-term relationships, potential conceptions, sociosexual behaviors, extra-pair sex, age at first sexual intercourse, and number of marital spouses.

- Reproductive domain: including both fertility and reproductive success, described below.

  i.   Fertility: number of children and grandchildren born, and age at the birth of the first child.
  ii.  Reproductive success: number of surviving children/grandchildren.

- Offspring mortality domain: mortality rate and number of deceased offspring.

Both published and unpublished studies were eligible. We restricted our sample to studies with adult participants ($\geq$ 17 years old). If key variables were collected but the relevant analyses were not reported, we contacted authors to request effect sizes or raw data. If data were reported in more than one study, we selected the analysis with the larger sample size or which included appropriate control variables, such as age. Studies using measures that were ambiguous and/or not comparable to measures used in other studies were excluded (e.g. measures of body size without information about the proportion of fat/muscle mass, or reproductive data during a very restricted time period). Twin studies where participants were sampled as pairs, population level studies, and studies analyzing both sexes together were also excluded, as well as articles that were not written in English or Swedish as we were not sufficiently fluent in other languages to conduct unbiased searching and extraction. Multiple measures from the same study were retained if they met the other criteria.

We chose Pearson's $r$ as our effect size measure and effect sizes not given as $r$ were converted (see *Supplementary file 1* for conversion formulas); if effect sizes were not convertible, the study was excluded. Where effect sizes for non-significant results were not stated in the article and could not be obtained, an effect size of 0 was assigned ($k = 28$). Excluding those effects from the analyses had no effect on any of the results. Twenty-nine percent of all observations (133 of 452, selected randomly) were double coded by the first author >2 months apart. Intracoder agreement was 97%. For coding decisions, see *Supplementary file 2*.

In total, 96 studies were selected (*Lassek and Gaulin, 2009*; *Hughes and Gallup, 2003*; *Weeden and Sabini, 2007*; *Frederick and Haselton, 2007*; *Lukaszewski et al., 2014*; *Hill et al., 2013*;

*Kordsmeyer et al., 2018*; *Peters et al., 2008*; *Boothroyd et al., 2017*; *Pawlowski et al., 2008*; *Arnocky et al., 2018*; *Rhodes et al., 2005*; *Van Dongen and Sprengers, 2012*; *Alvergne et al., 2009*; *Apicella, 2014*; *Apicella et al., 2007*; *Aronoff, 2017*; *Atkinson, 2012*; *Atkinson et al., 2012*; *Bogaert and Fisher, 1995*; *Booth et al., 1999*; *Boothroyd et al., 2011*; *Boothroyd et al., 2008*; *Charles and Alexander, 2011*; *Chaudhary et al., 2015*; *Edelstein et al., 2011*; *Falcon, 2016*; *Farrelly et al., 2015*; *Frederick, 2010*; *Frederick and Jenkins, 2015*; *Gallup et al., 2007*; *Genovese, 2008*; *Gettler et al., 2019*; *Gildner, 2018*;(*Gómez-Valdés et al., 2013*) *Hartl et al., 1982*; *Hoppler et al., 2018*; *Honekopp et al., 2007*; *Kirchengast, 2000*; *Kirchengast and Winkler, 1995*; *Klimas et al., 2019*; *Klimek et al., 2014*; *Kordsmeyer and Penke, 2017*; *Krzyżanowska et al., 2015*; *Kurzban and Weeden, 2005*; *Little et al., 1989*; *Loehr and O'Hara, 2013*; *Longman et al., 2018*; *Luevano et al., 2018*; *Maestripieri et al., 2014*; *Manning and Fink, 2008*; *Manning et al., 2003*; *Marczak et al., 2018*; *McIntyre et al., 2006*; *Međedović and Bulut, 2019*; *Mosing et al., 2015*; *Muller and Mazur, 1997*; *Nagelkerke et al., 2006*; *Nettle, 2002*; *Pawlowski et al., 2000*; *Pollet et al., 2011*; *Polo et al., 2019*; *Price et al., 2013*; *Prokop and Fedor, 2011*; *Prokop and Fedor, 2013*; *Puts et al., 2006*; *Puts et al., 2015*; *Putz et al., 2004*; *Rahman et al., 2005*; *Rosenfield et al., 2020*; *Schwarz et al., 2011*; *Scott and Bajema, 1982*; *Shoup and Gallup, 2008*; *Sim and Chun, 2016*; *Simmons and Roney, 2011*; *Smith et al., 2017*; *Sneade and Furnham, 2016*; *Sorokowski et al., 2013*; *Steiner, 2011*; *Stern et al., 2020*; *Strong, 2014*; *Strong and Luevano, 2014*; *Subramanian et al., 2009*; *Suire et al., 2018*; *Tao and Yin, 2016*; *van Anders et al., 2007*; *Varella et al., 2014*; *von Rueden et al., 2011*; *Voracek et al., 2010*; *Walther et al., 2016*; *Walther et al., 2017c*; *Walther et al., 2017a*; *Walther et al., 2017b*; *Waynforth, 1998*; *Winkler and Kirchengast, 1994*; *Honekopp et al., 2006*), comprising 474 effect sizes from 99 samples and 177,044 unique participants (*Table 1*). This exceeds the number of studies for each of the meta-analyses published previously (*Grebe et al., 2019*; *Van Dongen and Sprengers, 2012*; *von Rueden and Jaeggi, 2016*; *Xu et al., 2018*).

## Statistical analyses

We used the *metafor* package (*Viechtbauer, 2010*) in R 3.6.2 (*R Development Core Team, 2019*). *metafor* transforms Pearson's *r* to Fisher's *Z* for analysis; for ease of interpretation, effect sizes were converted back to *r* for presentation of results. For 2D:4D and voice pitch, effects were reverse coded prior to analysis because low values denote greater masculinity. Similarly, effects were reverse coded for all offspring mortality outcomes as well as the outcomes age at first birth and age at first sexual intercourse/contact, as low values denote increased fitness. In all analyses reported here, therefore, a positive value of *r* denotes a positive relationship between masculinity and fitness outcomes. All predicted relationships were positive.

Analyses were conducted using random-effects models, as we expected the true effect to vary across samples. We controlled for multiple comparisons by computing q-values (*Storey, 2002*). Note that q-values estimate the probability that a significant effect is truly significant or not; they are not adjusted p values. Thus, in all analyses presented below, only effects that remained significant after q-value computation (indicated by q-values < 0.05) are presented as significant. We computed q-values using all p values across all tests conducted in the whole analysis (266 in total). Q-values can be viewed in *Supplementary file 7*.

The analyses were conducted on three levels for both predictor traits and outcomes (*Figure 1*). For predictor traits, all six masculine traits were first combined and analyzed together at the *global masculinity level*. At the *trait level*, each masculine trait was then analyzed separately. Lastly, each masculine trait was further divided into separate *trait indices*, which were analyzed as potential moderators (see below).

For the outcomes, mating, reproduction, and offspring mortality were first analyzed together at the *total fitness level*. Given the widespread use of mating measures as proxies of reproductive outcomes, it is imperative where possible to test (and ideally compare) both mating and reproduction, to ensure that we are not relying on proxies that do not measure what they are assumed to measure. The *domain level* therefore divided outcomes into the *mating domain*, the *reproductive domain*, and the *offspring mortality domain* and analyzed them separately. The last level, the *measures level*, further divided mating and reproduction into their separate measures (mating attitudes and behaviors, and fertility and reproductive success, respectively), which were analyzed as subgroups.

**Table 1.** All studies included in the meta-analysis.

| Authors | Year | Predictor | Outcome | Sample | Sample location | Low or high fert. | N |
|---|---|---|---|---|---|---|---|
| *Alvergne et al., 2009* | 2009 | T | REP | Rural villagers | Senegal | High | 53 |
| *Apicella, 2014* | 2014 | Body masc | MAT, REP, OM | Hadza | Tanzania | High | 51 |
| *Apicella et al., 2007* | 2007 | Body masc, voice pitch, height | REP, OM | Hadza | Tanzania | High | 44-52 |
| *Arnocky et al., 2018* | 2018 | Facial masc | MAT | Students | Canada | Low | 135 |
| *Aronoff, 2017* | 2017 | T | MAT | Students | US | Low | 99 |
| *Atkinson, 2012* | 2012 | Body masc | MAT | Students | US | Low | 66 |
| *Atkinson et al., 2012* | 2012 | Body masc, voice pitch, height | REP | Himba (Ovahimba) | Namibia | High | 36 |
| *Bogaert and Fisher, 1995* | 1995 | T | MAT | Students | Canada | Low | 195-196 |
| *Booth et al., 1999* | 1999 | T | MAT | Army veterans and non-veterans | US | Low | 4393 |
| *Boothroyd et al., 2008* | 2008 | Facial masc | MAT | Students | UK | Low | 18-19 |
| *Boothroyd et al., 2011* | 2011 | Facial masc | MAT | Students | UK | Low | 36 |
| *Boothroyd et al., 2017* | 2017 | Facial masc | REP, OM | Agta | Philippines | High | 65 |
| | | Facial masc | MAT, REP, OM | Maya | Belize | High | 23-35 |
| *Charles and Alexander, 2011* | 2011 | 2D:4D, T | MAT | Students | US | Low | 25-42 |
| *Chaudhary et al., 2015* | 2015 | Body masc, height | MAT, REP, OM | Mbendjele BaYaka | Democratic Republic of the Congo | High | 55-73 |
| *Edelstein et al., 2011* | 2011 | T | MAT | Students | US | Low | 134 |
| *Falcon, 2016* | 2016 | 2D:4D | MAT | Students | US | Low | 137 |
| *Farrelly et al., 2015* | 2015 | T | MAT | Students | UK | Low | 75-78 |
| *Frederick, 2010* | 2010 | Body masc, 2D:4D, height | MAT | Students | US | Low | 61 |
| *Frederick and Haselton, 2007* | 2007 | Body masc | MAT | Students | US | Low | 56-121 |
| *Frederick and Jenkins, 2015* | 2015 | Height | MAT | Online | Worldwide | Low | 28759-31418 |
| *Gallup et al., 2007* | 2007 | Body masc, 2D:4D | MAT | Students | US | Low | 71-75 |
| *Genovese, 2008* | 2008 | Body masc | REP | Former teenage delinquents | US | High | 181 |
| *Gettler et al., 2019* | 2019 | T | MAT | Cebu Longitudinal Health and Nutrition Survey | Philippines | High | 288 |
| *Gildner, 2018* | 2018 | Body masc, 2D:4D, height | REP | Shuar Health and Life History Project | Ecuador | High | 48 |
| *Gómez-Valdés et al., 2013* | 2013 | Facial masc | REP | Hallstatt skulls | Austria | High | 179 |
| *Hartl et al., 1982* | 1982 | Body masc, height | MAT, REP | Former teenage delinquents | US | High | 180-185 |
| *Hill et al., 2013* | 2013 | Facial masc, body masc, voice pitch, height | MAT | Students | US | Low | 63 |
| *Hoppler et al., 2018* | 2018 | T | REP | Men's health 40+ study | Switzerland | Low | 268 |
| *Hughes and Gallup, 2003* | 2003 | Body masc | MAT | Students | US | Low | 50-59 |
| *Honekopp et al., 2006* | 2006 | 2D:4D, height | MAT | Students and non-students | Germany | Low | 79-99 |
| *Honekopp et al., 2007* | 2007 | Facial masc, body masc, height, T | MAT | Students and non-students | Germany | Low | 77 |
| *Kirchengast, 2000* | 2000 | Height | REP, OM | !Kung San | Namibia | High | 103 |

*Table 1 continued on next page*

*Table 1 continued*

| Authors | Year | Predictor | Outcome | Sample | Sample location | Low or high fert. | N |
|---|---|---|---|---|---|---|---|
| *Kirchengast and Winkler, 1995* | 1995 | Height | REP, OM | Urban and rural Kavango people | Namibia | High | 59-78 |
| *Klimas et al., 2019* | 2019 | T | MAT | Men's health 40+ study | Switzerland | Low | 159 |
| *Klimek et al., 2014* | 2014 | 2D:4D, height | REP | Mogielica Human Ecology Study Site | Poland | High | 238 |
| *Kordsmeyer et al., 2018* | 2018 | Body masc, voice pitch, height, T | MAT | Students and non-students | Germany | Low | 103-164 |
| *Kordsmeyer and Penke, 2017* | 2017 | 2D:4D, height | MAT | Students and non-students | Germany | Low | 141 |
| *Krzyżanowska et al., 2015* | 2015 | Height | REP | National Child Development Study | UK | Low | 6535 |
| *Kurzban and Weeden, 2005* | 2005 | Height | MAT, REP | Speed daters | US | Low | 1503-1501 |
| *Lassek and Gaulin, 2009* | 2009 | Body masc, height | MAT | NHANES III | US | Low | 4167-5159 |
| *Little et al., 1989* | 1989 | Height | REP, OM | Rural; growth stunted | Mexico | High | 103 |
| *Loehr and O'Hara, 2013* | 2013 | Facial masc | REP | WWII soldiers | Finland | High | 795 |
| *Longman et al., 2018* | 2018 | T | MAT | Students | UK | Low | 38 |
| *Luevano et al., 2018* | 2018 | Facial masc, height | MAT | Students | US | Low | 35-66 |
| *Lukaszewski et al., 2014* | 2014 | Body masc | MAT | Students | US | Low | 48-174 |
| *Maestripieri et al., 2014* | 2014 | T | MAT | Students | US | Low | 41-61 |
| *Manning and Fink, 2008* | 2008 | 2D:4D | MAT, REP | Online | Worldwide | Low | 26872-83681 |
| *Manning et al., 2003* | 2003 | 2D:4D | REP | Community | England | Low | 189 |
| | | 2D:4D | REP | Sugali and Yanadi tribal groups | India | High | 80 |
| | | 2D:4D | REP | Zulus from townships near Durban | South Africa | High | 66 |
| *Marczak et al., 2018* | 2018 | 2D:4D | REP | Yali | Indonesia | High | 47 |
| *McIntyre et al., 2006* | 2006 | T | MAT | Students | US | Low | 68-81 |
| *Međedović and Bulut, 2019* | 2019 | Height | MAT | Students | Serbia | Low | 39 |
| *Mosing et al., 2015* | 2015 | Height | MAT, REP | Study of Twin Adults: Genes and Environment | Sweden | Low | 2310-2549 |
| *Muller and Mazur, 1997* | 1997 | Facial masc | REP | West Point class of 1950 | US | High | 337 |
| *Nagelkerke et al., 2006* | 2006 | Height | MAT | NHANES 99–00 | US | Low | 798-809 |
| *Nettle, 2002* | 2002 | Height | REP | National Child Development Study | UK | Low | 4474 |
| *Pawlowski et al., 2008* | 2008 | Height | REP | Rural | Poland | High | 46 |
| *Pawlowski et al., 2000* | 2000 | Height | REP | Urban and rural | Poland | High | 3201 |
| *Peters et al., 2008* | 2008 | Facial masc, body masc, T | MAT | Students | Australia | Low | 100-113 |
| *Pollet et al., 2011* | 2011 | T | MAT | National Social Life, Health, and Aging Project | US | Low | 749 |
| *Polo et al., 2019* | 2019 | Facial masc, body masc, height | MAT | Students and non-students | Chile | Low | 198-206 |
| *Price et al., 2013* | 2013 | Body masc, height | MAT | Mainly students | UK | Low | 55 |
| *Prokop and Fedor, 2011* | 2011 | Height | REP | Friends and family of students | Slovakia | Low | 499 |
| *Prokop and Fedor, 2013* | 2013 | Height | MAT | Students | Slovakia | Low | 105-150 |

*Table 1 continued on next page*

*Table 1 continued*

| Authors | Year | Predictor | Outcome | Sample | Sample location | Low or high fert. | N |
|---|---|---|---|---|---|---|---|
| *Puts et al., 2006* | 2006 | Voice pitch | MAT | Students | US | Low | 103 |
| *Puts et al., 2015* | 2015 | T | MAT | Students | US | Low | 59-61 |
| *Putz et al., 2004* | 2004 | 2D:4D | MAT | Students | US | Low | 207-219 |
| *Rahman et al., 2005* | 2005 | 2D:4D, height | MAT | Students and non-students | UK | Low | 78-150 |
| *Rhodes et al., 2005* | 2005 | Facial masc, body masc, height | MAT | Mainly students | Australia | Low | 142-166 |
| *Rosenfield et al., 2020* | 2020 | Body masc, voice pitch, height | MAT, REP, OM | Tsimané | Bolivia | High | 55-62 |
| *Schwarz et al., 2011* | 2011 | 2D:4D | MAT | Students | Germany | Low | 52-89 |
| *Scott and Bajema, 1982* | 1982 | Height | REP | Third Harvard Growth Study | US | High | 606 |
| *Shoup and Gallup, 2008* | 2008 | Body masc, 2D:4D | MAT | Students | US | Low | 28-38 |
| *Sim and Chun, 2016* | 2016 | Body masc, 2D:4D | MAT | Students | US | Low | 90 |
| *Simmons and Roney, 2011* | 2011 | Body masc, T | MAT | Students | US | Low | 138 |
| *Smith et al., 2017* | 2017 | Body masc | REP | Hadza | Tanzania | High | 51 |
| *Sneade and Furnham, 2016* | 2016 | Body masc | MAT | Students | UK | Low | 145 |
| *Sorokowski et al., 2013* | 2013 | Height | REP, OM | Yali | Indonesia | High | 49-52 |
| *Steiner, 2011* | 2011 | 2D:4D, T | REP | Students and non-students | US | Low | 30 |
| *Stern et al., 2020* | 2020 | T | MAT | Students | UK | Low | 61 |
| *Strong, 2014* | 2014 | Body masc | MAT | Students | US | Low | 31 |
| *Strong and Luevano, 2014* | 2014 | Body masc, 2D:4D, height | MAT | Students | US | Low | 51-66 |
| *Subramanian et al., 2009* | 2009 | Height | OM | 2005-2006 National Family Health Survey | India | Low | 21120 |
| *Suire et al., 2018* | 2018 | Voice pitch | MAT | Mainly students | France | Low | 57-58 |
| *Tao and Yin, 2016* | 2016 | Height | REP | The Panel Study of Family Dynamics | Taiwan | Low | 1409 |
| *van Anders et al., 2007* | 2007 | T | MAT | Non-students | US | Low | 31 |
| *Van Dongen and Sprengers, 2012* | 2012 | Facial masc, body masc, 2D:4D | MAT | Not specified | Not specified | Low | 52 |
| *Varella et al., 2014* | 2014 | Body masc, 2D:4D, height | MAT | Students | Brazil, Czech Republic | Low | 69-80 |
| *von Rueden et al., 2011* | 2011 | Body masc, height | REP, OM | Tsimané | Bolivia | High | 162-197 |
| *Voracek et al., 2010* | 2010 | 2D:4D, height | REP | Firefighters | Austria | Low | 134 |
| *Walther et al., 2016* | 2016 | Body masc | REP | Men's health 40+ study | Switzerland | Low | 271 |
| *Walther et al., 2017a* | 2017a | Body masc | MAT | Men's health 40+ study | Switzerland | Low | 226 |
| *Walther et al., 2017b* | 2017b | Height | REP | Men's health 40+ study | Switzerland | Low | 271 |
| *Walther et al., 2017c* | 2017c | Height | MAT | Men's health 40+ study | Switzerland | Low | 226 |
| *Waynforth, 1998* | 1998 | 2D:4D, height | MAT, REP, OM | Villagers | Belize | High | 35-56 |
| *Weeden and Sabini, 2007* | 2007 | Body masc, 2D:4D, height | MAT | Students | US | Low | 188-212 |
| *Winkler and Kirchengast, 1994* | 1994 | Height | REP, OM | !Kung San | Namibia | High | 31-114 |

**Overall analyses (all traits & all samples combined)**

- Global masculinity (all masculine traits) predicting:
  - Total fitness
    - Mating domain
    - Reproductive domain
    - Offspring viability domain

**Main analyses (separating masculine traits; all samples combined)**

  - Each masculine trait predicting:
    - Mating domain
    - Reproductive domain

**Subgroup analyses (separating sample type & outcome measure type)**

- Low fertility samples:
  Each masculine trait predicting:
  - Mating domain
    - Mating attitudes
    - Mating behaviors
  - Reproductive domain
    - Fertility
    - Reproductive success
- High fertility samples:
  Each masculine trait predicting:
  - Mating domain
    - Mating attitudes
    - Mating behaviors
  - Reproductive domain
    - Fertility
    - Reproductive success

**Moderation analyses (full details in Supplementary Files 3)**

- Domain type (mating vs reproduction)
  - Mating measure type (attitudes vs behaviors)
  - Reproductive measure type (fertility vs reproductive success)
- Sample type (low vs high fertility)
  - Low fertility sample type (student vs non-student sample)
  - High fertility sample type (traditional vs industrialized sample)
- Ethnicity
- Marriage system
- Publication type
- Peer-review status
- Sexual orientation
- Transformation of variables
- Conversion of effect size
- Age control
- Other control variables

**Figure 1.** Overall analysis structure.

The mating domain comprised mating *attitudes* and mating *behaviors*, as high mating success may result from increased mating efforts (reflected in favorable attitudes towards short-term mating) and/or encountering more mating opportunities (reflected in mating behaviors) without actively seeking them (because of female choice, for example). It is therefore necessary to divide these two measures. The reproductive measures, fertility (number of offspring) and reproductive success (number of surviving offspring), are closely related but were also analyzed separately in subgroup analyses. Offspring mortality, on the other hand, was usually indexed by mortality *rate* (only two studies used

absolute number of dead offspring, and it made no difference to the results whether those studies were included or not) and is not directly related to offspring numbers. Offspring mortality was therefore analyzed as a separate domain. As there were too few observations of offspring mortality to test predictor traits separately, this outcome was only analyzed at the global masculinity level.

In addition to analyzing all samples together, we also analyzed low and high fertility samples separately to assess whether results were robust in both types of populations. We used a cut-off of three or more children per woman on average within that sample, which roughly corresponds to samples with vs without widespread access to contraception (*The World Bank, 2018*). Samples therefore had two levels: *all samples*, and the two sample types *low fertility* and *high fertility*.

The analysis structure was therefore as summarized in *Figure 1*: *overall analyses* tested global masculinity as a predictor of total fitness, as well as the three domains of mating, reproduction, and offspring mortality, separately, across all samples. In our *main analyses*, we analyzed masculinity at the trait level, in relation to the two outcome domains mating and reproduction. The following *subgroup analyses* considered low and high fertility samples separately, in addition to also dividing outcomes into their respective measures (mating attitudes vs mating behaviors, and fertility vs reproductive success).

Lastly, we performed a series of exploratory meta-regressions on potential moderator variables. Such moderation analyses compare effect sizes across categories of studies as determined by a particular study characteristic, for example monogamous vs polygynous marriage systems, to determine if effect sizes were robust and/or equivalent across these categories. Since power was often low, we ran moderation analyses separately for each study characteristic rather than trying to test for interactions. For all masculine traits where we had sufficient power, trait-general moderation analyses included: domain type (mating vs reproduction), mating measure type (attitudes vs behaviors), reproductive measure type (fertility vs reproductive success), sample type (low vs high fertility), low fertility sample type (student vs non-student), high fertility sample type (traditional vs industrialized), ethnicity, marriage system, publication status (published vs not published effect), peer review status (peer reviewed vs not peer reviewed), sexual orientation, transformation of variables, conversion of effect sizes, age control, and inclusion of other control variables. Note that since we included many non-published effects from studies that were published, 'publication status' referred to whether particular the particular *effects* were published, not the study as whole. The analysis can therefore detect evidence of any tendency for significant results to be 'written up' while nonsignificant ones are not, whether this bias occurs between or within manuscripts. We ran moderation analyses both for outcome domains and outcome measures (i.e. mating attitudes and mating behaviors, and fertility and reproductive success, respectively). For each masculine trait, we also conducted trait-specific moderation analyses (e.g. subjectively rated vs morphometric facial masculinity (for full details on trait-specific moderators, see *Supplementary file 3*)).

Analyses sometimes included more than one observation from the same study/sample. In all analyses, therefore, effect sizes were clustered both by sample and by study. For all analyses, only relationships with a minimum of three independent samples from a minimum of two separate studies were analyzed. For moderation analyses, this meant that *each category* of the moderator needed observations from at least three samples from at least two studies; in many cases, there were not enough observations to test for moderators.

In the Results section, unless otherwise specified, we summarize results from *trait-general* moderation analyses of outcome *domains* only (where results for outcome *measures* and *trait-specific* moderators are reported in *Supplementary file 4*). Additional details and full results of all analyses can be found in *Supplementary files 3-5*.

## Results
### Summary of samples

All 96 studies included in the meta-analysis are shown in *Table 1*. In total, 29 articles reported effect sizes from high fertility samples, which included 17 articles drawing on 13 different extant forager or subsistence populations (of the type sometimes referred to as 'small scale societies', coded here as non-industrialized) predominantly in Africa or Latin America. The remaining high fertility data came from historical samples or low socioeconomic status sub-populations within low-fertility countries (e.g.

agricultural Polish communities, former 'delinquents' in the US, and Zulus living in South African townships). Sixty-nine articles reported data from low fertility populations, which came from 54 primarily student or partially-student samples (43 of which were from English-speaking countries), and only 12 samples which could be considered representative community or cohort/panel samples. Two articles reported data drawn from 'global' online samples (classified as low fertility). The remaining low fertility samples were either unspecified or sampled particular sub-populations (e.g. specific professions).

## Overall analyses of global masculinity

In the initial overall analyses, global masculinity was weakly but significantly associated with greater total fitness (i.e. mating, reproduction, and offspring mortality combined) ($r$ = 0.080, 95% CI: [0.061, 0.101], $q$ = 0.001; we reiterate here that for all analyses, q-values < 0.05 denote significance after correcting for multiple comparisons). When we divided the outcome measures into their three domains, the positive (albeit weak) associations with global masculinity remained significant for mating, but not for reproduction or offspring mortality (mating: $r$ = 0.090, 95% CI: [0.071, 0.110], $q$ = 0.001; reproduction: $r$ = 0.047, 95% CI: [0.004, 0.090], $q$ = 0.080; offspring mortality: $r$ = .002, 95% CI: [–0.011, 0.015], $q$ = 0.475). While the effect was thus only significant for mating, the differences between effects were not significant, but we note that sample sizes differed considerably between domains.

Below, we present in further detail the results of the effect of global masculinity on each of the three outcome domains: mating, reproduction, and offspring mortality. We then present the associations between each masculine trait and mating and reproductive measures, separately. We also present results for subgroup and trait-general moderation analyses (for outcome domains only); for complete results, see *Supplementary files 4 and 5*.

## Mating

### Main analyses of each masculine trait

This set of analyses tested the prediction that individual masculine traits are positively associated with mating. In terms of the overall mating domain (i.e. mating attitudes and behaviors combined), all masculine traits showed the predicted positive relationships with mating, and the effects were significant for all traits except for facial masculinity and 2D:4D (*Table 2*). Some of these effects were very weak, however. The strongest associations with the mating domain were seen in terms of body masculinity ($r$ = 0.133, 95% CI: [0.091, 0.176], $q$ = 0.001; *Figure 2*), voice pitch ($r$ = 0.132, 95% CI: [0.061, 0.204], $q$ = 0.002; *Figure 3*), and testosterone levels ($r$ = 0.093, 95% CI: [0.066, 0.121], $q$ = 0.001; *Figure 4*). Height showed a significant but smaller effect size ($r$ = 0.057, 95% CI: [0.027, 0.087], $q$ = 0.002; *Figure 5*). While not the weakest association, the relationship between facial masculinity and mating was nonsignificant ($r$ = 0.080, 95% CI: [–0.003, 0.164], $q$ = 0.117). The effect for 2D:4D was also nonsignificant ($r$ = 0.034, 95% CI: [0.000, 0.069], $q$ = 0.102), and moderation analyses showed that this was the only trait that showed a significantly smaller effect size than the strongest predictor, body masculinity (p < 0.001, $q$ = 0.006).

### Comparison of high and low fertility samples

Across all masculine traits, most effect sizes (94%) came from low fertility samples. Moderation analyses of sample type could only be run for body masculinity and height; neither was significant, although in both cases, the effect sizes observed in the main analyses were significant only for low fertility, and not the less numerous high fertility samples ($k$ = 4 for each trait). The other four traits had only been measured in one high fertility sample each, and the main analyses thus contained almost exclusively low fertility samples. We further compared low fertility samples which were predominantly students with other low fertility samples as part of our moderation analyses where possible, that is for body masculinity, voice pitch, height, and testosterone. For body masculinity, student samples showed a significantly stronger effect than non-student samples for mating *behaviors* only ($B$ = –0.128, p = 0.009, $q$ = 0.032) but otherwise we found no differences (see *Supplementary file 4*).

### Inclusion bias/heterogeneity

Since the analysis included unpublished data, the distribution of effects in the funnel plots (see *Supplementary file 6A*) shows availability bias rather than publication bias. Apart from voice pitch, for which we did not have many effects, visual inspection of funnel plots indicated that they were generally

**Table 2.** Masculine traits predicting mating: main analyses and subgroup analyses of mating attitudes vs mating behaviors and low vs high fertility samples.

Pearson's r (95% CI); p value for meta-analytic effect, q-value (correcting for multiple comparisons); number of observations (k), samples (s), and unique participants (n); test for heterogeneity (Q), p value for heterogeneity. Statistically significant meta-analytic associations are bolded if still significant after controlling for multiple comparisons.

| | Mating | | | | | |
|---|---|---|---|---|---|---|
| Outcome: Sample | Facial masculinity | Body masculinity | 2D:4D | Voice pitch | Height | T levels |
| Mating domain: All samples | r = 0.080 (-0.003, 0.164), p = 0.060, q = 0.117 | **r = 0.133 (0.091, 0.176), p < 0.001, q = 0.001** | r = 0.034 (0.000, 0.069), p = 0.049, q = 0.102 | **r = 0.132 (0.061, 0.204), p < 0.001, q = 0.002** | **r = 0.057 (0.027, 0.087), p < 0.001, q = 0.002** | **r = 0.093 (0.066, 0.121), p < 0.001, q = 0.001** |
| | k = 30, s = 11, n = 948 | k = 121, s = 32, n = 7939 | k = 84, s = 23, n = 66,807 | k = 8, s = 5, n = 443 | k = 62, s = 25, n = 43,686 | k = 66, s = 21, n = 7083 |
| | Q(df = 29) = 54.834, p = 0.003 | Q(df = 120) = 297.472, p < 0.001 | Q(df = 83) = 101.994, p = 0.077 | Q(df = 7) = 2.334, p = 0.939 | Q(df = 61) = 263.247, p < 0.001 | Q(df = 65) = 66.090, p = 0.439 |
| Mating attitudes: All samples | r = .095 (-0.072, 0.263), p = 0.263, q = 0.304 | r = .078 (0.002, 0.155), p = 0.045, q = 0.098 | r = 0.035 (-0.061, 0.132), p = 0.474, q = 0.385 | s = 0 | r = 0.028 (-0.013, 0.068), p = 0.179, q = 0.253 | **r = 0.099 (0.026, 0.173), p = 0.008, q = 0.032** |
| | k = 5, s = 4, n = 407 | k = 20, s = 9, n = 922 | k = 19, s = 7, n = 504 | | k = 9, s = 6, n = 4232 | k = 21, s = 11, n = 1039 |
| | Q(df = 4) = 8.684, p = 0.070 | Q(df = 19) = 17.606, p = 0.549 | Q(df = 18) = 24.141, p = 0.151 | | Q(df = 8) = 5.137, p = 0.743 | Q(df = 20) = 25.379, p = 0.187 |
| Mating behaviors: All samples | r = .025 (-0.059, 0.109), p = 0.554, q = 0.424 | **r = .142 (0.099, 0.187), p < 0.001, q = 0.001** | r = 0.038 (-0.002, 0.078), p = 0.061, q = 0.117 | **r = 0.124 (0.043, 0.206), p = 0.003, q = 0.016** | **r = 0.054 (0.021, 0.087), p = 0.001, q = 0.008** | **r = 0.084 (0.058, 0.110), p < 0.001, q = 0.001** |
| | k = 22, s = 8, n = 755 | k = 91, s = 31, n = 7738 | k = 51, s = 19, n = 1607 | k = 7, s = 5, n = 443 | k = 48, s = 24, n = 42,179 | k = 32, s = 17, n = 6765 |
| | Q(df = 21) = 37.044, p = 0.017 | Q(df = 90) = 267.876, p < 0.001 | Q(df = 50) = 64.049, p = 0.087 | Q(df = 6) = 2.162, p = 0.904 | Q(df = 47) = 247.032, p < 0.001 | Q(df = 31) = 28.558, p = 0.592 |
| Mating domain: Low fert. samples | r = 0.089 (-0.001, 0.179), p = 0.053, q = 0.109 | **r = 0.135 (0.091, 0.180), p < 0.001, q = 0.001** | r = 0.038 (0.002, 0.073), p = 0.037, q = 0.086 | **r = 0.129 (0.055, 0.204), p < 0.001, q = 0.005** | **r = 0.055 (0.024, 0.086), p < 0.001, q = 0.004** | **r = 0.099 (0.069, 0.129), p < 0.001, q = 0.001** |
| | k = 28, s = 10, n = 913 | k = 117, s = 28, n = 7572 | k = 82, s = 22, n = 66,751 | k = 7, s = 4, n = 388 | k = 58, s = 21, n = 43,310 | k = 58, s = 20, n = 6795 |
| | Q(df = 27) = 54.287, p = 0.001 | Q(df = 116) = 289.080, p < 0.001 | Q(df = 81) = 101.369, p = 0.063 | Q(df = 6) = 2.234, p = 0.897 | Q(df = 57) = 259.576, p < 0.001 | Q(df = 57) = 61.443, p = 0.320 |
| Mating attitudes: Low fert. samples | r = 0.095 (-0.072, 0.262), p = 0.263, q = 0.304 | r = 0.078 (0.002, 0.155), p = 0.045, q = 0.098 | r = 0.035 (-0.061, 0.132), p = 0.474, q = 0.385 | s = 0 | r = 0.028 (-0.013, 0.068), p = 0.179, q = 0.253 | **r = 0.108 (0.021, 0.195), p = 0.015, q = 0.047** |
| | k = 5, s = 4, n = 407 | k = 20, s = 9, n = 922 | k = 19, s = 7, n = 504 | | k = 9, s = 6, n = 4,232 | k = 17, s = 10, n = 751 |
| | Q(df = 4) = 8.684, p = 0.070 | Q(df = 19) = 17.606, p = 0.549 | Q(df = 18) = 24.141, p = .151 | | Q(df = 8) = 5.137, p = 0.743 | Q(df = 16) = 20.017, p = 0.220 |
| Mating behaviors: Low fert. samples | r = 0.028 (-0.063, 0.119), p = 0.543, q = 0.420 | **r = 0.145 (0.100, 0.193), p< 0.001, q = 0.001** | r = 0.042 (0.001, 0.083), p = 0.045, q = 0.098 | **r = .119 (0.034, 0.205), p = 0.006, q = 0.025** | **r = .051 (0.017, 0.086), p = 0.004, q = 0.019** | **r = .088 (0.058, 0.119), p < 0.001, q = 0.001** |
| | k = 20, s = 7, n = 720 | k = 87, s = 27, n = 7371 | k = 49, s = 19, n = 1551 | k = 6, s = 4, n = 388 | k = 44, s = 20, n = 41,803 | k = 30, s = 16, n = 6477 |
| | Q(df = 19) = 36.610, p = 0.009 | Q(df = 86) = 259.448, p < 0.001 | Q(df = 48) = 62.941, p = 0.073 | Q(df = 5) = 2.017, p = 0.847 | Q(df = 43) = 243.392, p < 0.001 | Q(df = 29) = 27.793, p = 0.529 |
| Mating domain: High fert. samples | s = 1 | r = 0.105 (-0.069, 0.280), p = 0.235, q = 0.285 | s = 1 | s = 1 | r = 0.089 (-0.016, 0.193), p = 0.096, q = 0.157 | s = 1 |
| | | k = 4, s = 4, n = 367 | | | k = 4, s = 4, n = 376 | |
| | | Q(df = 3) = 7.282, p = 0.063 | | | Q(df = 3) = 3.388, p = 0.336 | |
| Mating attitudes: High fert. samples | s = 0 | s = 0 | s = 0 | s = 0 | s = 0 | s = 1 |
| Mating behaviors: High fert. samples | s = 1 | r = 0.105 (-0.069, 0.280), p = 0.235, q = 0.285 | s = 1 | s = 1 | r = 0.089 (-0.016, 0.193), p = 0.096, q = 0.157 | s = 1 |
| | | k = 4, s = 4, n = 367 | | | k = 4, s = 4, n = 376 | |
| | | Q(df = 3) = 7.282, p = 0.063 | | | Q(df = 3) = 3.388, p = 0.336 | |

*Note.* Fert. = fertility; k = number of observations; n = number of unique participants; Q = Cochran's Q test of heterogeneity; q = q-value; s = number of samples; T = testosterone.

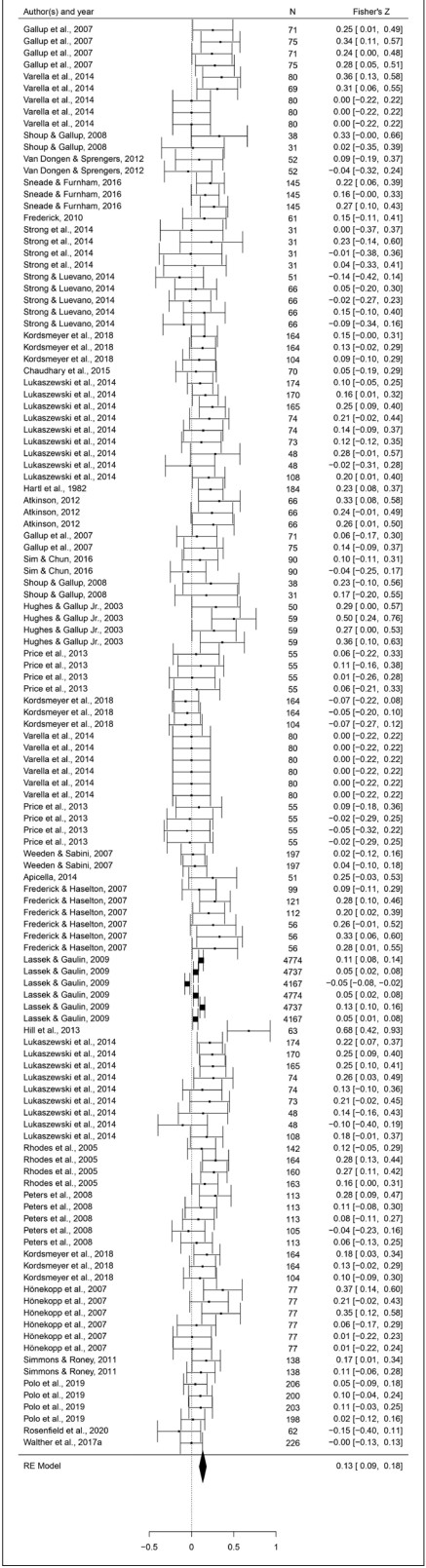

**Figure 2.** Forest plot of the association between body masculinity and the mating domain. Effect sizes are shown as *Z*-transformed *r*, with 95% confidence intervals in brackets. The width of the diamond

*Figure 2 continued*

corresponds to the confidence interval for the overall effect.

symmetric, suggesting that the analysis did not systematically lack studies with unexpected small effects. There was significant heterogeneity of effect sizes for facial masculinity, body masculinity, and height; all of which are accounted for in a random-effects analysis.

## Additional subgroup and moderation analyses for outcome domains

In this step of the analyses, we tested the hypothesis that each of the six masculine traits is positively associated with the two mating domain measures (mating attitudes and mating behaviors) and tested further potential control variables and trait-specific moderators. Results of subgroup analyses can be viewed in *Table 2* and trait-general moderators in Table 3; full results of all moderation analyses are reported in *Supplementary file 4*.

Type of mating measure (attitudes vs behaviors) was never a significant moderator. However, for both body masculinity and height, there were significant effects for mating behaviors (body masculinity: *r* = 0.142, 95% CI: [0.099, 0.187], *q* = 0.001, height: *r* = 0.054, 95% CI: [0.021, 0.087], *q* = 0.008) but not attitudes. Voice pitch was significantly related to mating behaviors (*r* = 0.124, 95% CI: [0.043, 0.206], *q* = 0.016) but was not measured in combination with mating attitudes. Testosterone levels showed near identical effects for both mating attitudes and behaviors (*r* = 0.099, 95% CI: [0.026, 0.173], *q* = 0.032 and *r* = 0.084, 95% CI: [0.058, 0.110], *q* = 0.001, respectively).

No trait-general moderator consistently changed the pattern of the associations (Table 3). Body masculinity effects were stronger in studies where age had not been controlled for compared to where it had been controlled for (*B* = 0.103, *p* = 0.015, *q* = 0.047). Associations for 2D:4D were weaker in non-white/mixed ethnicity samples compared to white samples (*B* = –0.080, p = 0.014, *q* = 0.047), and stronger where variables had been transformed to approximate normality compared to when they had not been transformed (*B* = 0.103, *p* = 0.016, *q* = 0.047). Similarly, associations for testosterone levels were also stronger for normality-transformed variables (*B* = 0.057, *p* = 0.015, *q* = 0.047), and weaker in gay/mixed sexuality samples compared to in heterosexual samples (*B* = –0.059, *p* = 0.003, *q* = 0.016).

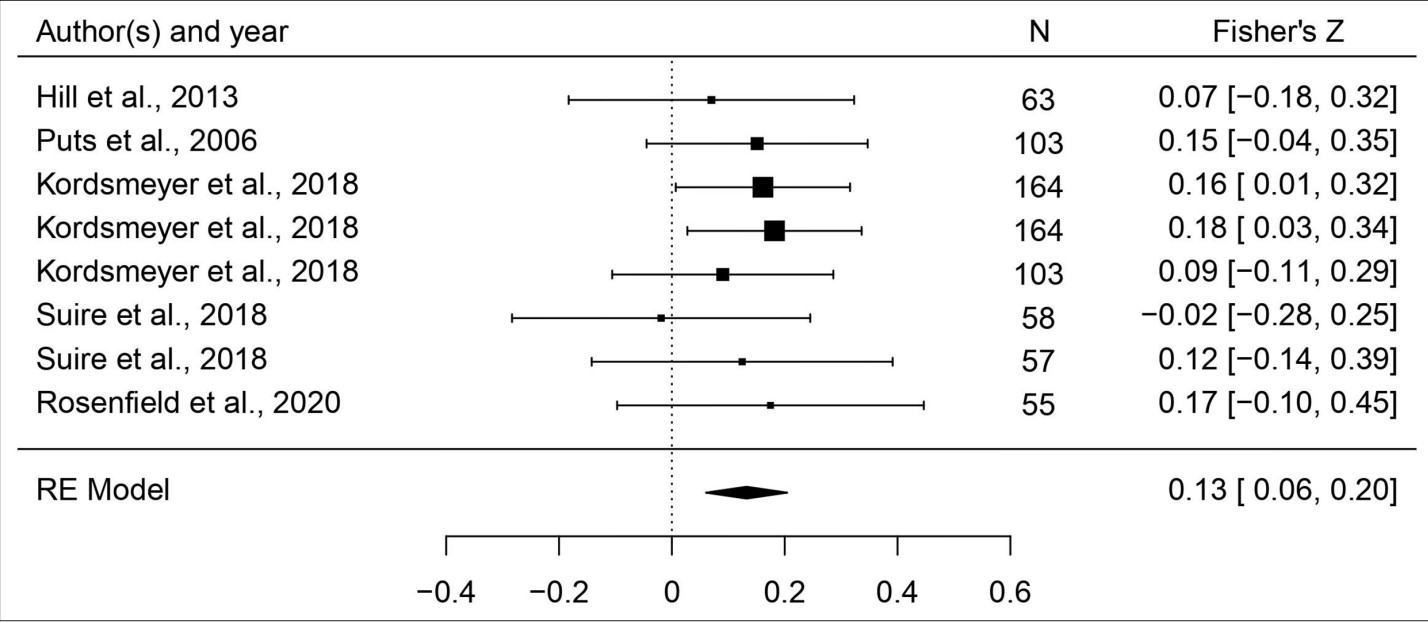

**Figure 3.** Forest plot of the association between voice pitch and the mating domain. Effect sizes are shown as Z-transformed *r*, with 95% confidence intervals in brackets. The width of the diamond corresponds to the confidence interval for the overall effect.

For trait-specific moderators, significant moderation was seen for *type* of body masculinity where body shape was a significantly weaker predictor than strength (B = –0.099, p = 0.003, q = 0.017). Effects for *rated* body masculinity were significantly stronger than for indices taken from body measurements (B = 0.177, p = 0.007, q = 0.029). For 2D:4D, studies that had measured digit ratios three times – rather than twice or an unknown number of times – showed significantly stronger effects (B = 0.102, p = 0.006, q = 0.025).

## Reproduction

### Main analyses of each masculine trait
In this set of analyses, we tested the hypothesis that individual masculine traits positively predict reproduction. As *Table 3* shows, relationships were generally in the predicted direction, but body masculinity was the strongest and only significant predictor (r = 0.143, 95% CI: [0.076, 0.212], q = 0.001; *Figure 6*). The only trait with an effect size significantly smaller than body masculinity was height (B = –0.107, p = 0.005, q = 0.023).

### Comparison of high and low fertility samples
The majority (77 %) of observations of reproduction were from high fertility samples. Moderation analyses of low versus high fertility samples could only be conducted for 2D:4D and height; effect sizes did not differ significantly between sample types. Comparing types of high fertility samples (industrialized vs non-industrialized) for 2D:4D and height did not show any differences in effect sizes (see *Supplementary file 4*). It was not possible to compare sample subtypes for the other traits because observations were almost entirely from non-industrialized populations.

### Inclusion bias/heterogeneity
Visual inspection of funnel plots (see *Supplementary file 6B*) suggested that while the effects for voice pitch, height, and testosterone levels were symmetrically distributed, our analysis may have lacked studies for the other traits. Facial masculinity and height showed significant heterogeneity.

### Additional subgroup and moderator analyses for outcome domains
Results of subgroup analyses can be viewed in *Table 3* and trait-general moderators in *Table 4*; full results of moderation analyses are found in *Supplementary file 4*.

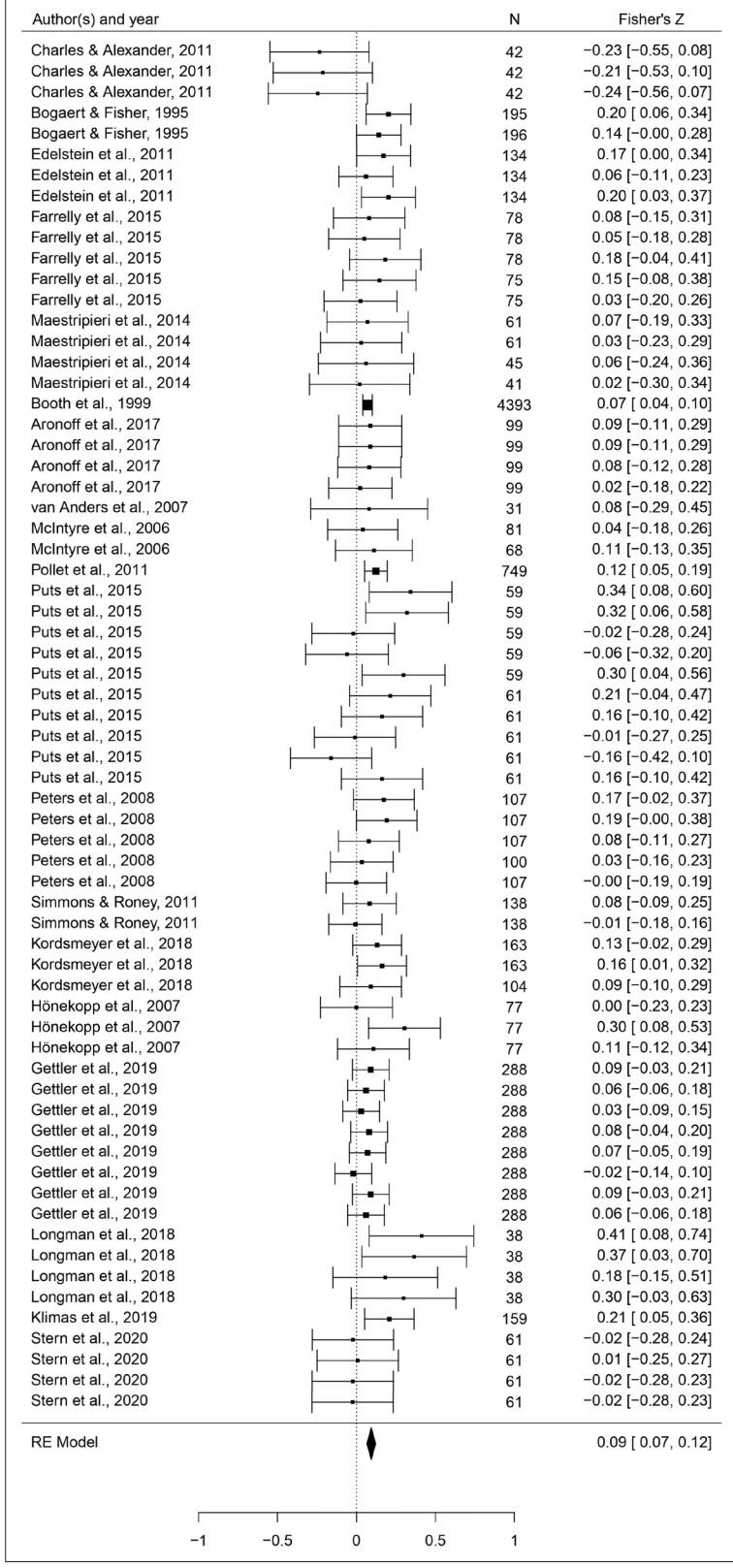

**Figure 4.** Forest plot of the association between testosterone levels and the mating domain. Effect sizes are shown as Z-transformed *r*, with 95% confidence intervals in brackets. The width of the diamond corresponds to the confidence interval for the overall effect.

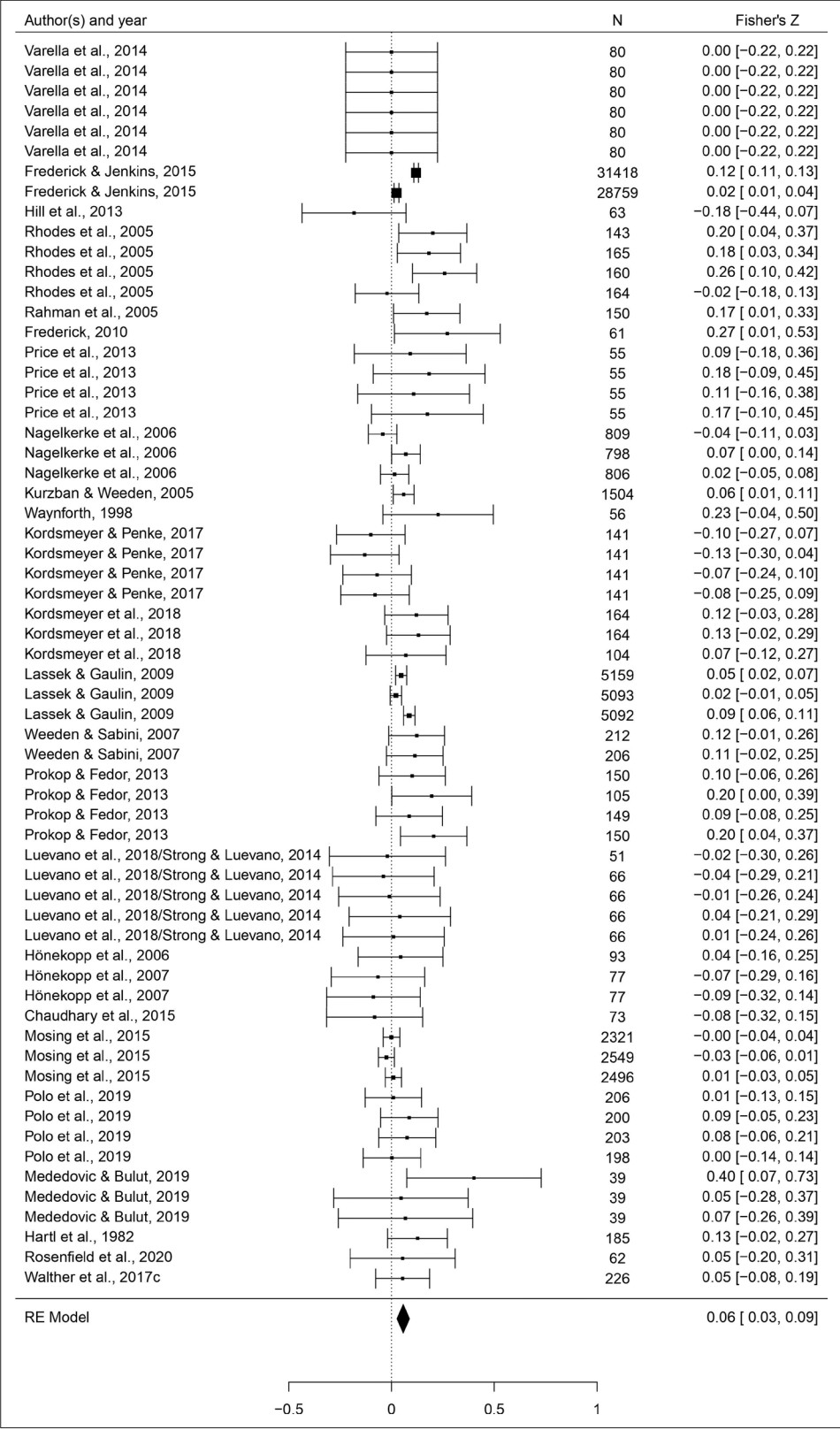

**Figure 5.** Forest plot of the association between height and the mating domain. Effect sizes are shown as *Z*-transformed *r*, with 95% confidence intervals in brackets. The width of the diamond corresponds to the confidence interval for the overall effect.

**Table 3.** Masculine traits predicting reproduction: main analyses and subgroup analyses of mating attitudes vs mating behaviors and low vs high fertility samples.

Pearson's r (95% CI); p value for meta-analytic effect, q-value (correcting for multiple comparisons); number of observations (k), samples (s), and unique participants (n); test for heterogeneity (Q), p value for heterogeneity. Statistically significant meta-analytic associations are bolded if still significant after controlling for multiple comparisons.

| Outcome: Sample | Reproduction | | | | | |
| --- | --- | --- | --- | --- | --- | --- |
| | Facial masculinity | Body masculinity | 2D:4D | Voice pitch | Height | T levels |
| Reproductive domain: All samples | $r = 0.099$ (-0.012, 0.211), p = 0.081, q = 0.140 | **$r = 0.143$ (0.076, 0.212), p < 0.001, q = 0.001** | $r = 0.074$ (-0.006, 0.154), p = 0.070, q = 0.131 | $r = 0.136$ (-0.053, 0.328), p = 0.158, q = 0.228 | $r = 0.006$ (-0.049, 0.062), p = 0.819, q = 0.491 | $r = 0.039$ (-0.067, 0.145), p = 0.474, q = 0.385 |
| | $k = 5, s = 5, n = 1411$ | $k = 14, s = 8, n = 897$ | $k = 19, s = 10, n = 84,558$ | $k = 5, s = 3, n = 143$ | $k = 35, s = 25, n = 22,326$ | $k = 3, s = 3, n = 351$ |
| | Q(df = 4) = 8.799, p = 0.066 | Q(df = 13) = 16.356, p = 0.230 | Q(df = 18) = 31.704, p = 0.024 | Q(df = 4) = 5.378, p = 0.251 | Q(df = 34) = 433.359, p < 0.001 | Q(df = 2) = 0.387, p = 0.824 |
| Fertility: All samples | $r = 0.003$ (-0.253, 0.260), p = 0.980, q = 0.543 | **$r = 0.130$ (0.060, 0.201), p < 0.001, q = 0.002** | $r = 0.032$ (-0.065, 0.130), p = 0.514, q = 0.406 | $s = 2$ | $r = 0.011$ (-0.039, 0.062), p = 0.660, q = 0.451 | $s = 2$ |
| | $k = 3, s = 3, n = 437$ | $k = 8, s = 6, n = 813$ | $k = 13, s = 5, n = 84,128$ | | $k = 26, s = 23, n = 22,242$ | |
| | Q(df = 2) = 5.416, p = 0.067 | Q(df = 7) = 4.840, p = 0.679 | Q(df = 12) = 17.757, p = 0.123 | | Q(df = 25) = 400.038, p < 0.001 | |
| RS: All samples | $s = 2$ | $r = 0.192$ (-0.052, 0.441), p = 0.122, q = 0.189 | **$r = 0.174$ (0.085, 0.267), p < 0.001, q = 0.002** | $s = 2$ | $r = -0.044$ (-0.201, 0.113), p = 0.584, q = 0.430 | $s = 1$ |
| | | $k = 6, s = 4, n = 205$ | $k = 6, s = 5, n = 430$ | | $k = 9, s = 9, n = 603$ | |
| | | Q(df = 5) = 11.344, p = 0.045 | Q(df = 5) = 0.976, p = 0.965 | | Q(df = 8) = 33.311, p < 0.001 | |
| Reproductive domain: Low fert. samples | $s = 0$ | $s = 1$ | $r = 0.083$ (-0.023, 0.190), p = 0.126, q = 0.191 | $s = 0$ | $r = -0.037$ (-0.112, 0.038), pp = 0.337, q = .347 | $s = 2$ |
| | | | $k = 8, s = 4, n = 84,034$ | | $k = 8, s = 8, n = 17,135$ | |
| | | | Q(df = 7) = 13.988, p = 0.051 | | Q(df = 7) = 244.970, p < 0.001 | |
| Fertility: Low fert. samples | $s = 0$ | $s = 1$ | $r = 0.052$ (-0.065, 0.169), p = 0.386, q = 0.369 | $s = 0$ | $r = -0.037$ (-0.112, 0.038), p = 0.337, q = 0.347 | $s = 2$ |
| | | | $k = 7, s = 3, n = 83,845$ | | $k = 8, s = 8, n = 17,135$ | |
| | | | Q(df = 6) = 8.335, p = 0.215 | | Q(df = 7) = 244.970, p < 0.001 | |
| RS: Low fert. samples | $s = 0$ | $s = 0$ | $s = 1$ | $s = 0$ | $s = 0$ | $s = 0$ |
| Reproductive domain: High fert. samples | $r = 0.099$ (-0.012, 0.211), p = 0.081, q = 0.140 | **$r = 0.163$ (0.104, 0.225), p < 0.001, q = 0.001** | $r = 0.083$ (-0.039, 0.205), p = 0.184, q = 0.257 | $r = 0.136$ (-0.053, 0.327), p = 0.158, q = 0.228 | $r = 0.034$ (-0.041, 0.109), p = 0.377, q = 0.367 | $s = 1$ |
| | $k = 5, s = 5, n = 1411$ | $k = 13, s = 7, n = 626$ | $k = 11, s = 6, n = 524$ | $k = 5, s = 3, n = 143$ | $k = 27, s = 17, n = 5191$ | |
| | Q(df = 4) = 8.799, p = 0.066 | Q(df = 12) = 12.347, p = 0.418 | Q(df = 10) = 12.595, p = 0.247 | Q(df = 4) = 5.378, p = 0.251 | Q(df = 26) = 70.216, p < 0.001 | |
| Fertility: High fert. samples | $r = 0.003$ (-0.253, 0.260), p = 0.980, q = 0.543 | **$r = 0.165$ (0.095, 0.237), p < 0.001, q = 0.001** | $s = 2$ | $s = 2$ | $r = 0.059$ (0.007, 0.111), p = 0.025, q = 0.068 | $s = 0$ |
| | $k = 3, s = 3, n = 437$ | $k = 7, s = 5, n = 542$ | | | $k = 18, s = 15, n = 5,107$ | |
| | Q(df = 2) = 5.416, p = 0.067 | Q(df = 6) = 0.988, p = 0.986 | | | Q(df = 17) = 26.458, p = 0.067 | |
| RS: High fert. samples | $s = 2$ | $r = 0.192$ (-0.052, 0.441), p = 0.122, q = 0.189 | **$r = 0.170$ (0.053, 0.291), p = 0.005, q = 0.022** | $s = 2$ | $r = -0.044$ (-0.201, 0.113), p = 0.584, q = 0.430 | $s = 1$ |
| | | $k = 6, s = 4, n = 205$ | $k = 5, s = 4, n = 241$ | | $k = 9, s = 9, n = 603$ | |
| | | Q(df = 5) = 11.344, p = 0.045 | Q(df = 4) = 0.965, p = 0.915 | | Q(df = 8) = 33.311, p < 0.001 | |

*Note.* fert. = fertility; k = number of observations; n = number of unique participants; Q = Cochran's Q test of heterogeneity; q = q-value; RS = reproductive success; s = number of samples; T = testosterone.

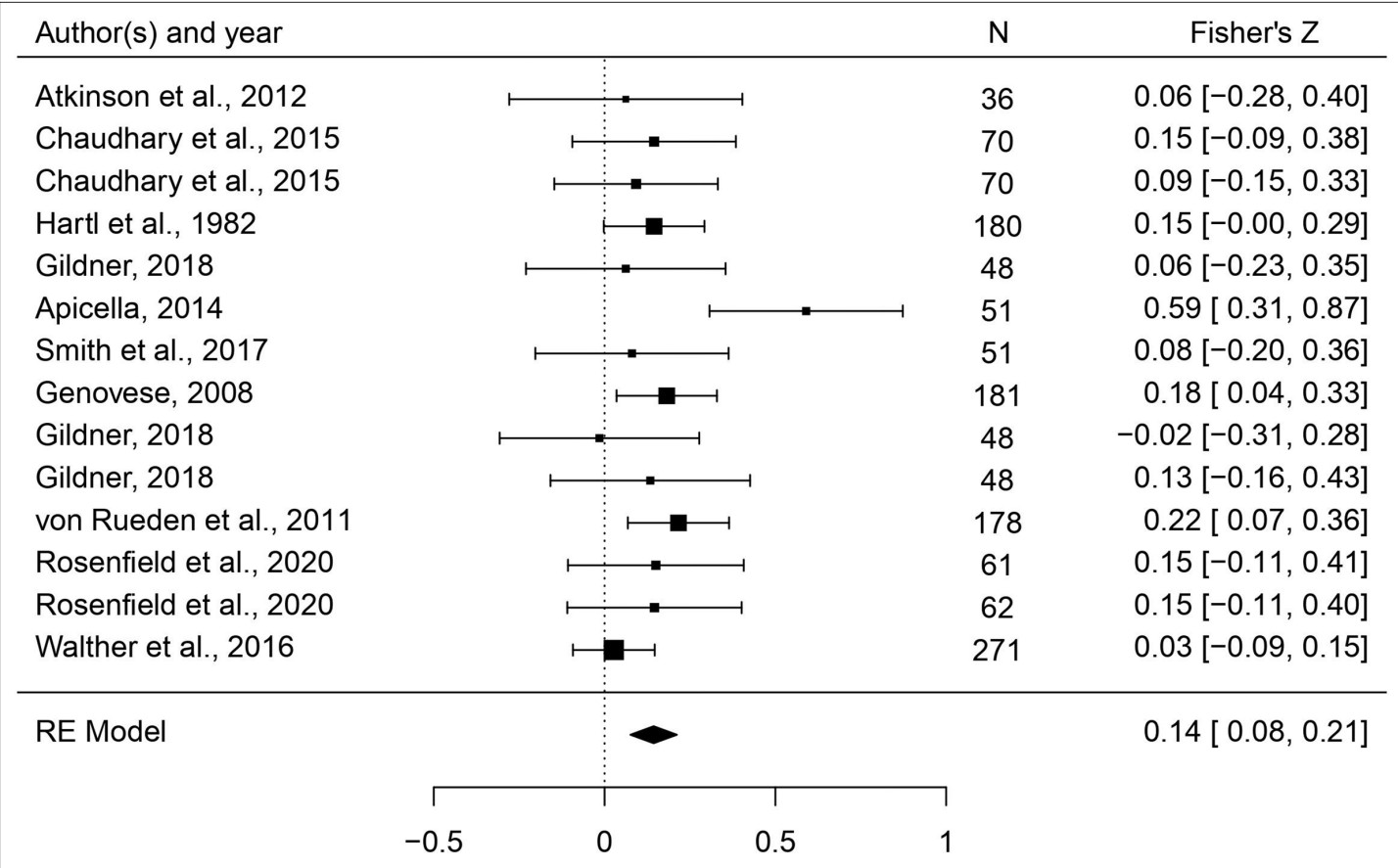

**Figure 6.** Forest plot of the association between body masculinity and the reproductive domain. Effect sizes are shown as Z-transformed *r*, with 95% confidence intervals in brackets. The width of the diamond corresponds to the confidence interval for the overall effect.

Moderation analyses (where possible) showed no evidence that the effects of masculinity traits on fertility differed from the effects on reproductive success. However, for body masculinity, the effect on fertility was significant (*r* = 0.130, 95% CI: [0.060, 0.201], *q* = 0.002; five out of six samples high fertility) while the somewhat larger effect on reproductive success was not. For 2D:4D, there was a significant effect for reproductive success (four out of five samples from high fertility populations: *r* = 0.174, 95% CI: [0.085, 0.267], *q* = 0.002) but not for fertility.

Similarly, for mating, no trait-general or trait-specific moderators had any consistent effects on the results. Body masculinity effects were stronger where effect sizes had been converted to Pearson's *r* compared to where they initially had been given as *r* (*B* = 0.143, p = 0.015, *q* = 0.047), and effects for height were stronger in gay/mixed sexuality samples than heterosexual samples (*B* = 0.135, p = 0.016, *q* = 0.047).

### Comparing mating and reproduction across traits
Moderation analyses of domain type (mating versus reproduction) for each trait showed no significant differences, although height and testosterone levels had weaker associations with reproduction than mating while body masculinity showed the opposite pattern. There were generally far fewer observations for reproductive measures, so this nonsignificant analysis may reflect lack of power. For facial masculinity, voice pitch, and 2D:4D, effect sizes for global mating and reproductive measures were near identical.

**Table 4.** Overview of moderation analyses for the mating vs reproductive domains.

Significant associations are indicated by+ and – signs, showing the direction of the moderator relative to the reference category (stated first in the moderator column); crosses indicate no significant moderation; and 'na' indicates that power was too low to run that specific analysis. Only associations that remained significant after controlling for multiple comparisons are indicated here. Note that this table only shows general moderators shared by all masculine traits; for trait-specific moderation analyses, see *Supplementary file 4*. Likewise, for moderation analyses of the two mating domain measures attitudes and behaviors, and the two reproductive domain measures fertility and reproductive success, we also refer to *Supplementary file 4*.

| | Facial masc. | | Body masc. | | 2D:4D | | Voice pitch | | Height | | T levels | |
|---|---|---|---|---|---|---|---|---|---|---|---|---|
| **Moderator** | **MAT** | **REP** | **MAT** | **REP** | **MAT** | **REP** | **MAT** | **REP** | **MAT** | **REP** | **MAT** | **REP** |
| Mating vs reproductive domain | X | | X | | X | | X | | X | | X | |
| Mating attitudes vs behaviors | X | na | X | na | X | na | na | na | X | na | X | na |
| Fertility vs reproductive success | na | na | na | X | na | X | na | na | na | X | na | na |
| Low vs high fertility sample | na | na | X | na | na | X | na | na | X | X | na | na |
| Low fertility: student vs non-student sample | na | na | X | na | X | na | na | na | X | na | X | na |
| High fertility: traditional vs industrialized sample | na | na | na | na | na | X | na | na | na | X | na | na |
| Predominantly white vs mixed/other/unknown ethnicity sample | X | na | X | na | – | X | na | na | X | X | X | na |
| Monogamous vs non-monogamous marriage system | na | na | X | na | na | X | na | na | na | X | na | na |
| Published vs non-published results | X | na | X | X | X | X | na | na | X | X | X | na |
| Peer reviewed vs not peer reviewed study | na | na | X | na | X | na | na | na | X | na | na | na |
| Heterosexual vs gay/mixed/unknown sample | X | na | X | na | X | X | na | na | X | + | – | na |
| Non-normality-transformed vs transformed variables | na | na | X | X | + | X | na | na | X | X | + | na |
| Non-converted vs converted effect sizes | na | na | X | + | na | na | na | na | X | X | X | na |
| Age controlled for vs not controlled for | X | na | + | na | X | X | na | na | X | X | X | na |
| Inclusion of non-relevant control variables vs not | na | na | na | X | na | na | na | na | na | X | X | na |

*Note.* Masc = masculinity; MAT = mating; REP = reproduction; T = testosterone.

# Discussion

## Summary of results

We conducted the first comprehensive meta-analysis of the relationships between men's masculine traits and outcomes related to mating and reproduction. Various proposed (and non-mutually exclusive) hypotheses suggest that more masculine men should show increased mating success (indexed by more matings and/or preferences for short-term mating), increased reproductive output (indexed by fertility and/or reproductive success), and/or lower offspring mortality. Our results showed partial support for these predictions. Global masculinity (i.e. all masculine traits combined) significantly predicted effects in the mating domain, but not the reproductive domain or the offspring mortality domain. When we analyzed each masculine trait separately, all traits except facial masculinity and 2D:4D significantly predicted effects in the mating domain, where similarly strong associations were seen for body masculinity, voice pitch, and testosterone levels, and a weaker correlation was seen for height. In terms of the reproductive domain, the only significant predictor was body masculinity. It was not possible to analyze offspring mortality at the specific predictor level owing to a severe lack of relevant data from which to draw conclusions (total number of observations for each outcome domain: mating domain $k = 371$; reproductive domain $k = 81$; offspring mortality domain $k = 22$).

We also examined how these effects play out in high versus low fertility populations. Typically, however, different outcomes were measured in different groups of populations; mating outcomes were predominantly measured in low fertility populations, while reproductive outcomes were measured mainly in high fertility populations. This made it more challenging to draw direct comparisons. Where it was possible to run moderation analyses on sample type, there were no significant differences. These analyses, however, have small numbers of high and low fertility samples in mating and reproductive outcomes respectively. Therefore, while we can confidently say that most forms of masculinity

(but not facial masculinity or 2D:4D) are associated with (largely self-reported) mating outcomes in low fertility samples, we cannot draw any clear conclusions regarding mating success in high fertility samples. Similarly, although we are confident that body masculinity is associated with fertility/reproductive success in high fertility samples, we cannot draw conclusions about low fertility contexts.

More generally, our moderation analyses on outcome types and factors relating to measure quality did not yield any consistent differences between effect sizes, suggesting that the effects we do find are reasonably robust within sample type at least. Two key points to note here are that: *i.* although effect sizes for mating attitudes and mating behaviors did differ for some traits (i.e. facial masculinity and body masculinity), these differences were never significant, despite mating behaviors being constrained by opportunities (assuming participants report truthfully), and *ii.* similarly, effect sizes did sometimes differ by publication status but never significantly so; in addition, the direction of the differences was not consistent (i.e. effect sizes were not consistently larger in published analyses). Even if the analysis was restricted to nonpublished effects only, the association between body masculinity and both mating and reproduction would be weaker but remain significant (mating: $r = 0.077$, $p = 0.006$; reproduction: $r = 0.112$, $p < 0.001$; both associations would remain significant after q-value computation). Overall, this suggests that researchers have not been selectively reporting larger effect sizes.

Compared to previous meta-analyses assessing associations between handgrip strength and mating outcomes (*Van Dongen and Sprengers, 2012*), height/strength and reproductive outcomes (*von Rueden and Jaeggi, 2016*; *Xu et al., 2018*), and testosterone levels and mating effort (*Grebe et al., 2019*), our analysis benefits from more comprehensive measures of masculinity, larger sample sizes, and inclusion of more unpublished effects. With the exception of Xu and colleagues' analysis (*Xu et al., 2018*), we observe smaller effect sizes than previous meta-analyses, which suggests that the association between masculinity and fitness outcomes has previously been overestimated. In general, what significant associations we did observe were small and ranged between $r = 0.05$ and $0.17$, although they are potentially meaningful in an evolutionary context. As benchmarks for interpreting correlations, *Funder and Ozer, 2019* suggest that a correlation of 0.10, while being a small effect, has the potential to be influential over a long time period, and a medium-size correlation of 0.20 can be consequential both in the short- and long-term. The cumulative effect of relatively 'weak' correlations can therefore be of real consequence, particularly when considered in terms of selection acting over many generations.

## Major implications

### Selection for body masculinity

The first stand-out result of our analysis is that body masculinity (i.e. strength/muscularity) is the only trait in our analysis that was consistently correlated with both mating and reproductive outcomes across populations, and the effects of body masculinity on these outcomes were among the strongest in the analysis. In contrast, other aspects of masculinity (except facial masculinity and 2D:4D) predicted mating success in low fertility samples but did not yield reproductive benefits in high fertility samples.

Body masculinity is therefore the trait where we have the most compelling evidence that selection is currently happening within naturally fertile populations - and from that, can infer that selection likely took place in prior eras as well. As such, our results are consistent with the argument that dimorphisms in strength and muscle mass are sexually selected. Overall, since traits such as body size, strength, and muscularity are associated with formidability, our findings are consistent with the male-male competition hypothesis. In species with male intrasexual competition, males tend to evolve to become larger, stronger, and more formidable than females, as they are in humans. Some authors argue that male-male violence has influenced human evolution (*Hill et al., 2016*; *Gat, 2015*), and male intergroup aggression increases mating/reproductive success in both non-industrialized human societies and in non-human primates (*Glowacki and Wrangham, 2015*; *Manson et al., 1991*). (And indeed the non-human evidence might suggest this form of dimorphism has been under selection since prehominid ancestors, although the strength of such selection pressures have likely fluctuated over this time [172].) For example, in the Yanomamö Indians, men who kill others have greater reproductive success (*Chagnon, 1988*). A relationship between formidable traits and fitness outcomes need not be a direct one, however. It might, as mentioned in the introduction, be mediated by other factors that are important in mate choice, such as interpersonal status and dominance. For example, features that are advantageous in intraspecies conflicts may also be advantageous when hunting game (*Sell et al.,*

*2012*; *Smith et al., 2017*) reported that in a hunter-gatherer population, men with greater upper body strength and a low voice pitch had increased reproductive success, but this relationship was explained by hunting reputation.

It is of course possible that different selection pressures may have contributed to the evolution of different masculine traits. Male-male competition for resources and mates, female choice, and inter-group violence are all plausible, non-mutually exclusive explanations (*Plavcan, 2012*). In this article, we have focused on the effect of men's own traits on their fitness, but it is of course equally possible that men varying in masculinity may differ in the quality of the mates they acquire. If masculine men are able to secure mates who are more fertile and/or better parents, this may also increase their fitness.

## No evidence of advantage for facial masculinity

Considerable attention has been given in the literature to the hypothesis that masculinity in men's facial structure is an indicator of heritable immunocompetence (i.e. good genes), which should then be associated with greater mating and reproductive success. While we find that the effect of facial masculinity on mating was similar in size to that of other traits ($r = 0.08$), it was not significantly different from zero, suggesting more variability in effects. Furthermore, the effect of facial masculinity on mating (such as it was) was largely driven by mating attitudes and was close to zero for mating behaviors, suggesting that men's facial masculinity exerts virtually no influence on mating when moderated by female choice. Similarly, the influence of facial masculinity on fertility in high fertility samples was non-existent ($r = 0.00$). Although the relationship with reproductive success appeared stronger, this was based on only two samples. This is, all together, doubly striking because although voice pitch, height, and testosterone levels did not predict reproductive outcomes, they did all relate to mating in the expected direction. Facial masculinity is ergo an outlier in being so entirely unrelated to mating success in our data, while subject to so large a literature assuming the opposite.

Overall, these findings contradict a large body of literature claiming that women's preferences for masculinity in men's faces are adaptive. Rather, they indicate that such preferences (to the extent they exist at all) are a modern anomaly only found in industrialized populations, as suggested by Scott and colleagues (*Scott et al., 2014*), and as demonstrated by the positive correlation between facial masculinity preferences and national health and human development indices (*Marcinkowska et al., 2019*).

## Students and foragers

One key observation regarding our dataset is that it shows a rather 'bimodal' distribution between a large number of studies sampling (predominantly English-speaking) students on one hand, and a cluster of studies sampling foragers, horticulturalists, and other subsistence farmers (predominantly from just two continents) on the other. Where it was possible to compare student vs non-student/mixed samples within low fertility populations, and traditional vs industrialized high fertility samples, we generally did not find any differences. Likewise, where it was possible to compare monogamous and formally polygynous cultures, we also found no differences. This is despite evidence that monogamy actually changes selection pressures on human men (*Brown et al., 2009*). Therefore, although we are reasonably confident that our results regarding body masculinity and reproduction are robust, insofar as they are based on non-industrialized populations with a range of subsistence patterns (hunter-gatherers, forager-horticulturalists, and pastoralists), it remains essential to consider rebalancing the literature. Not only do we require more holistic representation of non-industrialized populations (drawing from Asia and Oceania in particular, where we had one and zero samples, respectively), but it is also important to increase representation of non-student participants in low fertility contexts.

## Disconnection between mating and reproductive literatures

As noted above, we found that voice pitch, height, and testosterone levels were associated with (largely self-reported) mating success in mostly low fertility populations, but not with actual reproductive fitness in high fertility populations. A caveat here is that effect sizes for voice pitch and reproduction were similar in strength to effect sizes for body masculinity, but we note that this analysis had the smallest sample size of our whole analysis ($k = 5$, $n = 143$), which prevents us from drawing firm conclusions regarding the relationship between voice pitch and reproductive outcomes.

Overall, however, the contradicting pattern of results for the traits mentioned above raise important concerns for the human sexual selection field, particularly with respect to whether (and which) mating measures can be used as reliable indicators of likely ancestral fitness when considering the current evidence base. Since reproductive outcomes – for good reason – are not considered meaningful fitness measures in populations with widespread contraception use, we typically test fitness outcomes in industrialized populations using mating measures such as sociosexual attitudes and casual sexual encounters. This is done under the assumption that such measures index mating strategies that ancestrally would have increased men's offspring numbers. However, if mating outcomes (be it attitudinal or behavioral) measured in low fertility populations truly index reproductive outcomes in naturally fertile contexts, we would expect traits that predict mating to also predict reproduction on average across samples (notwithstanding the diversity in norms/reproductive behaviors across high fertility samples). We do not, however, have evidence that this is generally the case. Our findings therefore raise the question of whether these widely used measurements are truly valid proxies of what we purport to be measuring.

Our findings thus illustrate that when we attempt to test the same underlying research questions using different measurements in different populations, this may yield conclusions that are erroneous or misleading when applied outside of the studied population. We suggest, based on our analysis, that researchers could for instance consistently gather sexual partner number, age of marriage, and number/survival rates of offspring in multiple population types. Wherever possible, it is essential to use the same measurements across populations, or at least resist the temptation of applying our findings universally.

## Key limitations

### Non-linearity

A limitation of our analysis is that we only assessed linear relationships, ignoring possible curvilinear associations. There is evidence suggesting that moderate levels of masculinity might be associated with increased reproductive success (see e.g. *Boothroyd et al., 2017*, for offspring survival rates) and perceived attractiveness (*Frederick and Haselton, 2007*; *Johnston et al., 2001*, but see also *Sell et al., 2017*), with a decrease for both very low and very high levels of masculinity. Indeed, some of these authors have argued that masculinity may be under stabilizing, rather than directional, selection in humans. In instances such as these, our 'null' conclusions regarding e.g. facial masculinity, remain valid; facial masculinity does not appear to be under directional selection. However, we also note that there is data suggesting that height in men may be optimal when it is over-average but not maximal. In this scenario, although the linear relationship would be weaker, the trait remains under directional selection, and we would still expect to see positive, albeit weak, associations in our analyses. In the vast majority of studies included, only linear relationships were tested, and acquiring original data to investigate and synthesize non-linear effects was beyond the scope of the current article. However, increased publication of open data with articles may well facilitate such a project in future years.

### Testosterone effects

As mentioned above, in our analysis testosterone levels predicted mating outcomes – with similar effect sizes for attitudinal and behavioral measures – but did not predict reproduction. While a causal relationship between testosterone levels and mating success cannot be established from this (i.e. whether high testosterone men pursue more mating opportunities which leads to more matings, or whether high testosterone results from many matings), testosterone is commonly argued to motivate investment in mating effort. *If* current testosterone levels index degree of masculine trait expression in men, our results *might* indicate that masculine men's increased mating success is due to greater pursuit of matings - rather than reflecting female choice and/or greater competitiveness. Two caveats for interpreting our results, however (applicable both to the significant effect we observe for mating and the nonsignificant effect for reproduction), is that circulating testosterone levels *i.* change over the course of a man's lifetime, peaking in early adulthood and subsequently declining (*Booth and Dabbs, 1993*, although this may not be the case in non-industrialized populations: *Bribiescas, 1996*), and *ii.* are reactive. In the studies we gathered, testosterone levels were generally measured contemporaneously with mating/reproductive data collection – not when masculine traits generally become exaggerated in adolescence. Testosterone also decreases, for example, when men enter a relationship

or get married (*Archer, 2006*; *Holmboe et al., 2017*), when they become fathers (*Archer, 2006*; *Gettler et al., 2011*), or when they engage in childcare (*Archer, 2006*). Thus, men whose testosterone levels were previously high may show declining testosterone levels either because of their age and/or because their relationship or fatherhood status has changed. This limits the conclusions we can draw, both with regards to a potential mediating role of testosterone levels in the association between masculine traits and mating success, and the observed nonexistent effect for testosterone levels and reproductive outcomes. We also note that the sample size for reproduction, as a function of testosterone levels, was small.

## Conclusion

In summary, we used a large-scale meta-analysis of six masculine traits and their relationships with mating and reproductive outcomes to test whether such traits are currently under selection in humans. We found that all masculine traits except facial masculinity and 2D:4D were associated with significantly greater mating success. However, only body masculinity predicted higher fertility, indexed by reproductive onset, number of offspring, and grand-offspring. We further note that the mating and reproduction literature is starkly split between studying mating in predominantly student settings, and 'only' fertility in high fertility settings, which imposes constraints on both this paper and our field as a whole. We argue that our findings illustrate that when we test hypotheses about human evolution largely in industrialized populations, we risk drawing conclusions that are not supported outside of evolutionarily novel, highly niche mating and reproductive contexts. We therefore call for greater sample diversity and more homogenous measurements in future research.

## Additional information

### Funding
No external funding was received for this work.

### Author contributions
Linda H Lidborg, Conceptualization, Data curation, Formal analysis, Investigation, Methodology, Project administration, Visualization, Writing - original draft, Writing - review and editing; Catharine Penelope Cross, Conceptualization, Methodology, Writing - review and editing; Lynda G Boothroyd, Conceptualization, Formal analysis, Investigation, Methodology, Supervision, Writing - review and editing

### Author ORCIDs
Linda H Lidborg ⓘ http://orcid.org/0000-0001-9667-9326
Catharine Penelope Cross ⓘ http://orcid.org/0000-0001-8110-8408
Lynda G Boothroyd ⓘ http://orcid.org/0000-0001-6660-5828

### Decision letter and Author response
Decision letter https://doi.org/10.7554/eLife.65031.sa1
Author response https://doi.org/10.7554/eLife.65031.sa2

## Additional files

### Supplementary files
- Supplementary file 1. Effect size conversion formulas.
- Supplementary file 2. Study coding decisions.
- Supplementary file 3. Description of moderators.
- Supplementary file 4. Moderation analyses.
- Supplementary file 5. Global masculinity analyses.
- Supplementary file 6. Funnel plots of effect sizes for mating measures.
- Supplementary file 7. Output for q-value computation for all analyses.

• Transparent reporting form

## Data availability

All data generated and analysed in this article, including complete R code, are available on the Open Science Framework ( https://doi.org/10.17605/OSF.IO/PHC4X).

The following dataset was generated:

| Author(s) | Year | Dataset title | Dataset URL | Database and Identifier |
|---|---|---|---|---|
| Lidborg LH, Cross CP, Boothroyd LG | 2020 | Is male dimorphism under sexual selection in humans? A meta-analysis | https://doi.org/10.17605/OSF.IO/PHC4X | Open Science Framework, 10.17605/OSF.IO/PHC4X |

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
