## [Editor Report]

This paper presents a series of meta-analyses to test the plausibility of sexual selection hypotheses for the origins and/or maintenance of six sexually differentiated traits in humans, with strength/muscularity found to be significantly associated with both mating and reproduction. The authors considered both published and unpublished datasets to help account for potential publication biases that could theoretically impact meta-analysis results, which is a particular strength of this study.

---

## [Decision Letter]

**Decision letter after peer review:**

Thank you for submitting your article "Is male dimorphism under sexual selection in humans? A meta-analysis" for consideration by *eLife*. Your article has been reviewed by 4 peer reviewers, one of whom is a member of our Board of Reviewing Editors, and the evaluation has been overseen by George Perry as the Senior Editor. The following individuals involved in review of your submission have agreed to reveal their identity: Jordan Martin (Reviewer #2); Steven Gangestad (Reviewer #3); Urszula Marcinkowska (Reviewer #4).

The reviewers included experts from both evolutionary psychology and evolutionary anthropology, as these two approaches do not always overlap in readership. I include a list of items below that summarizes the main issues raised by the reviewers, all of whom were in agreement that this was a strong, well-written paper that would benefit from revision. However, the recommended revisions do permeate the entire manuscript.

Summary Items

1. Framing and introduction: The introduction should be scaled back, to arrive more quickly at the analysis part of the paper. However, this scaling should not just be a cutting process, but instead involves the most important suggested revision of the paper. In this, the authors should situate the study within a more robust theoretical framing of the problem, drawing more strongly especially from evolutionary anthropology. The reason for this is linked to the next issue, which is the degree to which we should consider the study populations in the meta-analysis to be representative of humans in general. The authors provide an excellent review, but it could be shortened a bit and exchanged for the theoretical piece. This would shift in the framing of the paper to be more of an exploration of the different patterns of data reported in this field, with more skepticism about reporting bias, rather than mainly a primary hypothesis test. The authors here have a terrific opportunity in this compiled dataset to examine if these hypotheses are actually testable (or to what extent) using available data. For example, do the observations between X and X trait hold true when excluding studies presenting this as their main hypothesis? In this way, their paper becomes more than just another manuscript of many that attempts to test extant hypotheses, and more of a critique and technical examination of the state of available data – which would be ultimately more useful across a range of disciplines. As a structural note, the authors could also divide the introduction and Discussion sections into subsections, for example when providing different hypothesis behind masculinity attractiveness.

2. All reviewers suggested caution that the data in most of the analyses came from WEIRD populations, and that in specific realms (especially child mortality and marriage patterns) there is likely to be strong cultural variation that can bias patterns away from a stronger ancestral signal. The authors are clearly aware of this, but it is an issue used to explain results, rather than to frame expectations. In the detailed comments we each provide specific places where the authors might temper their discussion to specifically address this issue. They should also be more up front at the start in addressing how representative the meta-analysis based on these populations is likely to be of humans in general (or, indeed, humans over evolutionary time). The authors do perform an assessment of biased reporting of results, but most of this appears in the discussion and it is difficult to know if it is possible to show, using this dataset, if there is actually bias in the underlying data. If the authors reframe the manuscript as an examination of the robustness of existing data, then they can better consider the limitations of their sample and meta-analysis for informing historical selection pressures on human populations, as well as for identifying the direct and indirect targets of sexual selection. This is closely related to the limited scope of the theoretical framework presented for interpreting the data.

3. The authors performed many, many statistical tests in this study, but did not perform any multiple test correction or estimate of false discovery rates. They then highlight individual tests with p-values lower than 0.05, but this is expected many times with such an analysis scheme. The authors should provide a sense of which (if any) tests are relatively unexpected by chance when considering all of these tests performed. One option would be to compute FDR q-values with the input being all p-values from all analyses conducted as part of the study (they could alternatively choose to be more conservative than this and use Bonferroni correction). This could then focus the paper on the most robust results (if any remain standing), which links to items #1 and #2 about reframing the manuscript in terms of the utility of current data to speak to the issue of male dimorphism, and shifting more emphasis to some of the potential biases inherent in the datasets.

Line by line items and individual figure and table requests

– Line 29: Higher variance in male reproduction and a surplus of reproductively available males seems to be a true statement mainly with respect to mammals. The references are about humans and non-human primates, so this should be specified here.

– Line 42: I suggest underlining caution when interpreting results of 2d:4d and sexual reproduction studies, as very often they were not replicated (as authors note by adding reference 15).

– Line 60: as the previous sentence is a long one (but sufficiently clear, so no need to shorten it) I would maybe rephrase the "…such putative associations…" to reiterate which associations the authors meant.

– Line 67: "better quality, more viable offspring." as I learnt from experience, readers from fields other than human biology/ecology often are put off by saying offspring can be of better or worse quality. I suggest adding a short description, of what does a good quality offspring mean.

– Line 69: shouldn't there be "what" instead of "which"?

– Line 79: although there are citations here, I still wonder about the logic that masculinity should be linked to "health", broadly construed. There are many reasons why poor "health" might be linked to masculinity, for example through lifestyle choices or the social context of masculine performance. I would appreciate more nuance here. Are there specifically studies that examine "health" and masculinity in which lifestyle and diet, for example, are comparable? If those cited here qualify, then that should be specified.

– Line 80: authors could add here some of the cross-cultural studies showing women's mixed preferences for facial masculinity (DeBruine et al., 2010, 2011, Brooks et al., 2011 or Marcinkowska et al., 2019).

– Line 104: I suggest splitting the paragraph around line 104. The previous part describes the sexy-sons hypothesis, while sentences after line 104 focus on other explanations and possible negative correlates of pronounced sexual dimorphism in men. I also believe that explaining more in-depth the trade-off that women face when judging sexual dimorphism could be useful in the introduction (now there are just two sentences on the putative negative correlates of masculinity).

– Line 123: "In societies without effective contraception, reproduction can be measured directly in terms of offspring numbers and/or offspring survival." I understand the logic here, but at the end of the day everything about human behavior is culturally mediated. Humans understand the relationship between sex and children, and so contraception is only part of the equation; if it is societally acceptable or encouraged to have children, then more children will be had – irrespective of the availability of contraception. Similarly, if there are reasons not to have children but contraception is not available, humans deploy cultural means to moderate fertility. Many societies are also in the midst of a demographic transition, as these ideas (and access to contraception) are in the midst of changing across one generation to the next. Furthermore, how representative are "high fertility" and "low fertility" populations, if there is so much underlying cultural and individual variation in reproductive choices? These are all just thoughts I had while reading the paper that could potentially feed into an expanded discussion that draws in more work from evolutionary anthropology.

– Line 141: "Lastly, a meta-analysis of 16 effects by Grebe et al., (57) showed that men with high testosterone levels invested more in mating effort, indexed by mating with more partners and showing greater interest in casual sex (r = 0.22)."

I appreciate the authors' comments on confounding within- vs between-individual T effects on mating behavior in the discussion, but it would also be useful here to emphasize that these findings are for circulating levels of testosterone, to avoid readers assuming this study clearly demonstrated between-individual effects.

– Line 142: "testosterone levels" measured how/when?

– Line 155: "We make no attempt to evaluate the hypotheses described above against each other."

This statement should probably be deleted as it seems contradicted by the claim in the discussion that "our findings could be interpreted as lending support to the male-male competition hypothesis".

– Line 207: "Studies using measures that were ambiguous and/or not comparable to measures used in other studies were excluded." Please provide an example of what such a measure would be.

– Line 214: "Where effect sizes for non-significant results were not stated in the paper and could not be obtained, an effect size of 0 was assigned." Please state here how many studies were affected by this decision.

– Line 273: "We used a cut-off of three or more children per woman on average within that sample, which roughly corresponds to samples with vs without widespread access to contraception." It would be helpful to include a raw data plot in the supplement showing this pattern visually to better justify the analytic decision. My apologies if I happened to miss this.

– Line 275: a reference could be useful here to back up the cut-off point if one exists.

– Line 281: "high" is missing before fertility.

– Line 296: authors could provide possible options for" publication status" and "peer-review status"

– Line 299: calling this variable of a study "quality" infers that one approach is better than the other (while I don't think that is the case).

– Line 395: "For 2D:4D, all samples except one were from low fertility populations. Although the relationship with the mating domain was significant (but very weak), this was no longer significant when excluding non-significant effect sizes not reported in the paper, where we had assigned an effect size of 0." Is this just a power issue?

– Line 494: missing r for the weak association.

– Line 495: would it be possible to add here a number/rate/percentage showing that there was much more missing data in the "offspring mortality" analyses than there was in other models?

– Line 555: and also visible in the positive correlation of facial masculinity preference with Health and HDI Indexes found in Marcinkowska et al., 2019.

– Line 556: "A limitation of our analysis is that we only assessed linear relationships, ignoring possible curvilinear associations… However, if such results indicate that greater-than-average levels of masculinity are associated with peak fitness/attractiveness, we would still expect to see positive, albeit weak, linear relationships." I sincerely appreciate the authors making this point in their discussion. However, they should also consider that some populations may be at an evolutionary equilibrium with regard to the measured traits, in which case we would expect null main effects accompanied by quadratic effects (i.e. stabilizing selection). In general, more attention is needed to selection on not only the mean phenotype within a population, but also its variance.

– Lande, R., and Arnold, S. J. (1983). The measurement of selection on correlated characters. Evolution, 1210-1226.

– Stinchcombe, J. R., Agrawal, A. F., Hohenlohe, P. A., Arnold, S. J., and Blows, M. W. (2008). Estimating nonlinear selection gradients using quadratic regression coefficients: double or nothing? Evolution, 62(9), 2435-2440.

– In a paragraph starting at 563 authors could add how the T was measured in the listed studies, to further back up the idea that current T might not be related to current masculinity.

– Line 588: "Wherever possible, we thus need to use the same measurements across populations, or at least resist the temptation of applying our findings universally."

– Line 589: maybe here authors could add what these variables could be according to their exhaustive analysis.

– Line 595: "higher fertility." as measured by …

– Line 598: typo, should be "evolutionarily". Also, great point. Also, the reproductive context is niche (would there be as much casual sex in a population where it is more likely to lead to offspring?), not just the mating context.

Figures

– Figure 1: This is a very helpful figure for the reader. I would consider using a slightly smaller indentation, though. Please also note that the figure is quite low resolution in the main text, if you were not already aware.

– Figure 3: This forest plot is very helpful. I would encourage the authors to include more forest plots for other meaningful effects in the paper, as they will be of much greater relevance to the typical reader than the multiple, large funnel plots, which are important but could just as well be placed in the supplement and referenced from the main text.

Tables

– I am curious why Table 1 and 2 are presented as separate tables. It is a very long section in this article, and structuring it as clear as possible will be very helpful.

– Table 5: Please bold the checks to make them easier to see. Currently, the Xs stand out much more than the checks. Also, I would consider using a different color scheme to aid colorblind readers, as well as including the direction of effect change (i.e. + or -) next to the checks with the reference categories indicated in the footnote.

– Is there a reason why the authors did not discuss the moderation effects from Table 5 in the Discussion section?

[Editors’ note: further revisions were suggested prior to acceptance, as described below.]

Thank you for resubmitting your work entitled "A meta-analysis of the association between male dimorphism and fitness outcomes in humans" for further consideration by *eLife*. Your revised article has been evaluated by George Perry (Senior Editor) and a Reviewing Editor.

While I do find that your manuscript has been improved and you have done some solid work in addressing many reviewer comments, I (speaking as a Senior Editor) have some concerns on several points that I do not feel were properly addressed, and yet that I consider fundamental. I give an overview of these comments in the following, and we offer you the opportunity to address these comments robustly in a revision, or to consider submitting your manuscript to another journal. In addition, several comments shared by the Reviewing Editor who handled your manuscript, Jessica Thompson, are below.

Senior Editor comments:

1. This is a passage from essential revision #1 from the previous decision letter:

"This would shift in the framing of the paper to be more of an exploration of the different patterns of data reported in this field, with more skepticism about reporting bias, rather than mainly a primary hypothesis test. The authors here have a terrific opportunity in this compiled dataset to examine if these hypotheses are actually testable (or to what extent) using available data. For example, do the observations between X and X trait hold true when excluding studies presenting this as their main hypothesis? In this way, their paper becomes more than just another manuscript of many that attempts to test extant hypotheses, and more of a critique and technical examination of the state of available data – which would be ultimately more useful across a range of disciplines."

In my view, two of the most essential parts of this point was not accomplished in the revision. One, for a shift to a different type of paper that one necessarily aiming to draw biological conclusions. Two, for skepticism related to reporting bias (i.e. the tendency for negative results to not be published, or even for the effects of inadvertent or worse p-hacking in the published literature, coming into the meta-analyses).

2. Related to FDR q-values. Thank you for providing these in the revision. But how was this analysis performed? It is surprising that the q-values are as low as they are being reported, considering the large number of tests performed. I think that the most conservative approach should be used here (all p-values from all of the tests in the paper), given the implications and the large number of tests performed.

---

## [Author Response]

Summary Items1. Framing and introduction: The introduction should be scaled back, to arrive more quickly at the analysis part of the paper. However, this scaling should not just be a cutting process, but instead involves the most important suggested revision of the paper. In this, the authors should situate the study within a more robust theoretical framing of the problem, drawing more strongly especially from evolutionary anthropology. The reason for this is linked to the next issue, which is the degree to which we should consider the study populations in the meta-analysis to be representative of humans in general. The authors provide an excellent review, but it could be shortened a bit and exchanged for the theoretical piece. This would shift in the framing of the paper to be more of an exploration of the different patterns of data reported in this field, with more skepticism about reporting bias, rather than mainly a primary hypothesis test. The authors here have a terrific opportunity in this compiled dataset to examine if these hypotheses are actually testable (or to what extent) using available data. For example, do the observations between X and X trait hold true when excluding studies presenting this as their main hypothesis? In this way, their paper becomes more than just another manuscript of many that attempts to test extant hypotheses, and more of a critique and technical examination of the state of available data – which would be ultimately more useful across a range of disciplines. As a structural note, the authors could also divide the introduction and Discussion sections into subsections, for example when providing different hypothesis behind masculinity attractiveness.2. All reviewers suggested caution that the data in most of the analyses came from WEIRD populations, and that in specific realms (especially child mortality and marriage patterns) there is likely to be strong cultural variation that can bias patterns away from a stronger ancestral signal. The authors are clearly aware of this, but it is an issue used to explain results, rather than to frame expectations. In the detailed comments we each provide specific places where the authors might temper their discussion to specifically address this issue. They should also be more up front at the start in addressing how representative the meta-analysis based on these populations is likely to be of humans in general (or, indeed, humans over evolutionary time). The authors do perform an assessment of biased reporting of results, but most of this appears in the discussion and it is difficult to know if it is possible to show, using this dataset, if there is actually bias in the underlying data. If the authors reframe the manuscript as an examination of the robustness of existing data, then they can better consider the limitations of their sample and meta-analysis for informing historical selection pressures on human populations, as well as for identifying the direct and indirect targets of sexual selection. This is closely related to the limited scope of the theoretical framework presented for interpreting the data.

We address points 1 and 2 together as they both concern framing and focus of the paper. We thank the reviewers for raising the key point around the importance of discussing what the data *can* actually show, and the need to consider diversity (or lack of) within both high and low fertility samples. Based on their suggestions above and line by line comments below we have done the following:

Added more subheadings throughout

– Trimmed down the theoretical discussion in the introduction to make space for other points

– Expanded on the diversity in low fertility samples in the introduction

– Clarified the importance in considering how predictions regarding adaptations would play out in low vs high fertility samples in the introduction and explicitly reporting the high vs low fertility sample results in their own sub-sections within the results. We also create an explicit focus on this point and the problem with lacking mating data in high fertility samples in particular, in the discussion.

– More explicitly noted our analysis of publication status in the introduction in order to comment on the results (showing no difference in effects reported as an observed result in papers vs data noted but results not reported, or data not even published.)

– Added a summary of sample types in the Results, which breaks down samples beyond the simply high and low fertility distinction, and discussed the implications in Discussion.

Regarding the suggestion made by some reviewers that we are unduly accepting of our overall results (and their largely null direction), we would note that although one author (LB) is a known long-standing sceptic of the immunocompetence hypothesis, all authors considered it plausible at the start of this investigation that positive associations would be found for most traits with both mating success and fertility and our conclusions are based entirely on the results we found. Furthermore, our high fertility samples are in fact more diverse than the low fertility data (which is largely students) and as such the null results regarding most traits and fertility are arguably the more convincing.

We believe that we have now produced a paper which answers both the question we began with (is there evidence that masculine traits predict enhanced RS and/or mating success) and which considers more carefully the challenges arising from structural imbalances in the literature regarding the tendency in our field to collect data from either students *or* African and Latin American ‘small scale societies’, while largely ignoring other populations.

We believe these changes have improved the paper and hope that they will prove satisfactory.

3. The authors performed many, many statistical tests in this study, but did not perform any multiple test correction or estimate of false discovery rates. They then highlight individual tests with p-values lower than 0.05, but this is expected many times with such an analysis scheme. The authors should provide a sense of which (if any) tests are relatively unexpected by chance when considering all of these tests performed. One option would be to compute FDR q-values with the input being all p-values from all analyses conducted as part of the study (they could alternatively choose to be more conservative than this and use Bonferroni correction). This could then focus the paper on the most robust results (if any remain standing), which links to items #1 and #2 about reframing the manuscript in terms of the utility of current data to speak to the issue of male dimorphism, and shifting more emphasis to some of the potential biases inherent in the datasets.

We have used q values as the reviewers have suggested and focused on the remaining significant results. The overall pattern of results does not change much; the relationship between 2D:4D and mating success is no longer significant, but body masculinity remains the key trait associated with both mating and reproductive outcomes.

Line by line items and individual figure and table requests– Line 29: Higher variance in male reproduction and a surplus of reproductively available males seems to be a true statement mainly with respect to mammals. The references are about humans and non-human primates, so this should be specified here.

We now specify this refers to humans and non-human primates (now on line 30).

– Line 42: I suggest underlining caution when interpreting results of 2d:4d and sexual reproduction studies, as very often they were not replicated (as authors note by adding reference 15).

Do the reviewers mean sexual dimorphism here? If so, we agree and thus cited reference 14 (line 43).

– Line 60: as the previous sentence is a long one (but sufficiently clear, so no need to shorten it) I would maybe rephrase the "…such putative associations…" to reiterate which associations the authors meant.

This phrase has now gone due to trimmed theoretical sections.

– Line 67: "better quality, more viable offspring." as I learnt from experience, readers from fields other than human biology/ecology often are put off by saying offspring can be of better or worse quality. I suggest adding a short description, of what does a good quality offspring mean.

We appreciate this sensitivity check and have replaced ‘better quality, more viable offspring’ with ‘healthier and more viable offspring, who are more likely to survive’ (line 63).

– Line 69: shouldn't there be "what" instead of "which"?

This phrase has now gone due to trimmed theoretical sections.

– Line 79: although there are citations here, I still wonder about the logic that masculinity should be linked to "health", broadly construed. There are many reasons why poor "health" might be linked to masculinity, for example through lifestyle choices or the social context of masculine performance. I would appreciate more nuance here. Are there specifically studies that examine "health" and masculinity in which lifestyle and diet, for example, are comparable? If those cited here qualify, then that should be specified.

We agree with the reviewer that the proposed association between masculinity and health is very complex, and it is indeed a claim frequently contested in the literature. We have expanded on that point in our reworked introduction. Our aim here, however, is not to evaluate the evidence for or against this claim, but rather to describe it as one of the mechanisms that have been proposed as an explanation of how selection may act on masculine traits, whilst acknowledging to the reader that it is indeed contested. As such, this passage is intentionally brief (lines 72-74).

– Line 80: authors could add here some of the cross-cultural studies showing women's mixed preferences for facial masculinity (DeBruine et al., 2010, 2011, Brooks et al., 2011 or Marcinkowska et al., 2019).

See the previous comment (lines 72-74). This point, with references, is also raised in the discussion (lines 614-618).

– Line 104: I suggest splitting the paragraph around line 104. The previous part describes the sexy-sons hypothesis, while sentences after line 104 focus on other explanations and possible negative correlates of pronounced sexual dimorphism in men. I also believe that explaining more in-depth the trade-off that women face when judging sexual dimorphism could be useful in the introduction (now there are just two sentences on the putative negative correlates of masculinity).

We have split this paragraph after ‘which should in turn result in more offspring.’ And have briefly expanded on the quality-care trade off (lines 94-107).

– Line 123: "In societies without effective contraception, reproduction can be measured directly in terms of offspring numbers and/or offspring survival." I understand the logic here, but at the end of the day everything about human behavior is culturally mediated. Humans understand the relationship between sex and children, and so contraception is only part of the equation; if it is societally acceptable or encouraged to have children, then more children will be had – irrespective of the availability of contraception. Similarly, if there are reasons not to have children but contraception is not available, humans deploy cultural means to moderate fertility. Many societies are also in the midst of a demographic transition, as these ideas (and access to contraception) are in the midst of changing across one generation to the next. Furthermore, how representative are "high fertility" and "low fertility" populations, if there is so much underlying cultural and individual variation in reproductive choices? These are all just thoughts I had while reading the paper that could potentially feed into an expanded discussion that draws in more work from evolutionary anthropology.

We have now expanded on the diversity of ‘high fertility’/non-contracepting populations in the introduction (lines 108-143; see also our first reply above), and have deleted this sentence.

– Line 141: "Lastly, a meta-analysis of 16 effects by Grebe et al., (57) showed that men with high testosterone levels invested more in mating effort, indexed by mating with more partners and showing greater interest in casual sex (r = 0.22)."I appreciate the authors' comments on confounding within- vs between-individual T effects on mating behavior in the discussion, but it would also be useful here to emphasize that these findings are for circulating levels of testosterone, to avoid readers assuming this study clearly demonstrated between-individual effects.

We’ve added that the Grebe et al., study used circulating testosterone (lines 159-160).

– Line 142: "testosterone levels" measured how/when?

We now specify that testosterone was assayed by blood or saliva (line 160). The authors of the paper mention that sampling time of day was recorded, but this information is not given anywhere.

– Line 155: "We make no attempt to evaluate the hypotheses described above against each other."This statement should probably be deleted as it seems contradicted by the claim in the discussion that "our findings could be interpreted as lending support to the male-male competition hypothesis".

We have cut this as part of our rewritten introduction more explicitly focusing on whether the effects exist, at all, in high v low fertility populations.

– Line 207: "Studies using measures that were ambiguous and/or not comparable to measures used in other studies were excluded." Please provide an example of what such a measure would be.

We have added examples of such measures (lines 234-236).

– Line 214: "Where effect sizes for non-significant results were not stated in the paper and could not be obtained, an effect size of 0 was assigned." Please state here how many studies were affected by this decision.

This has now been stated (line 243).

– Line 273: "We used a cut-off of three or more children per woman on average within that sample, which roughly corresponds to samples with vs without widespread access to contraception." It would be helpful to include a raw data plot in the supplement showing this pattern visually to better justify the analytic decision. My apologies if I happened to miss this.

The rationale for the choice of this cut-off (which was made prior to data collection/analysis) was to distinguish between populations with and without widespread access to contraception; the global fertility rate is around 2.5, and most industrialized countries have a fertility rate around or below 2 children/woman, whereas natural fertility populations typically have a fertility rate around 4 or higher (according to the World Bank Data: https://data.worldbank.org/indicator/SP.DYN.TFRT.IN). Additionally, industrialized, contracepting populations typically show a decline from a fertility rate above or around 3 to between 1.5-2 when contraception become widely available. A cut-off of 3 children/woman was therefore judged as appropriate to distinguish between low and high fertility populations. We have added a reference to justify this decision (line 302).

– Line 275: a reference could be useful here to back up the cut-off point if one exists.

We have added a reference the World Bank data also cited above (line 302).

– Line 281: "high" is missing before fertility.

This has been added (line 309).

– Line 296: authors could provide possible options for" publication status" and "peer-review status"

This has been added, along with a clarification that publication status refers to publication status of a particular effect, not status of the study it was taken from (lines 324-329).

– Line 299: calling this variable of a study "quality" infers that one approach is better than the other (while I don't think that is the case).

We have deleted this.

– Line 395: "For 2D:4D, all samples except one were from low fertility populations. Although the relationship with the mating domain was significant (but very weak), this was no longer significant when excluding non-significant effect sizes not reported in the paper, where we had assigned an effect size of 0." Is this just a power issue?

Yes – including the assigned effect sizes of.0, the association is just significant (*p* = 0.049), thus becomes nonsignificant when those effects are excluded. However, this association does not survive corrections for multiple comparisons (using q values as suggested) so this difference is no longer relevant.

– Line 494: missing r for the weak association.

This association did not survive correction for multiple comparisons.

– Line 495: would it be possible to add here a number/rate/percentage showing that there was much more missing data in the "offspring mortality" analyses than there was in other models?

We have added total number of observations for each outcome domain (lines 528-529).

– Line 555: and also visible in the positive correlation of facial masculinity preference with Health and HDI Indexes found in Marcinkowska et al., 2019.

This has been added (lines 617-618).

– Line 556: “A limitation of our analysis is that we only assessed linear relationships, ignoring possible curvilinear associations… However, if such results indicate that greater-than-average levels of masculinity are associated with peak fitness/attractiveness, we would still expect to see positive, albeit weak, linear relationships.” I sincerely appreciate the authors making this point in their discussion. However, they should also consider that some populations may be at an evolutionary equilibrium with regard to the measured traits, in which case we would expect null main effects accompanied by quadratic effects (i.e. stabilizing selection).

We thank the reviewers for raising this point. We were indeed referring in part to research which found such quadratic effects regarding masculinity and offspring quality and have made the point more clearly now, as well as differentiating stabilizing selection from non-linear but still directional selection (as some have argued is the case for height). We have now expanded on this point (668-682).

– In a paragraph starting at 563 authors could add how the T was measured in the listed studies, to further back up the idea that current T might not be related to current masculinity.

We have noted this in order to make the point clearer that T levels were generally gathered in a manner which exacerbates the reactive T problem: “In the studies we gathered, testosterone levels were generally measured contemporaneously with mating/reproductive data collection.” (lines 697-698).

– Line 588: “Wherever possible, we thus need to use the same measurements across populations, or at least resist the temptation of applying our findings universally.”– Line 589: maybe here authors could add what these variables could be according to their exhaustive analysis.

We have added a preceding sentence: “We suggest, based on our analysis, that researchers could for instance consistently gather sexual partner number, age of marriage, and number/survival rates of offspring in multiple population types.” (lines 686-688).

– Line 595: “higher fertility.” As measured by …

This has been added (lines 713-714).

– Line 598: typo, should be “evolutionarily”. Also, great point. Also, the reproductive context is niche (would there be as much casual sex in a population where it is more likely to lead to offspring?), not just the mating context.

Typo corrected; ‘reproductive contexts’ added (line 719).

Figures– Figure 1: This is a very helpful figure for the reader. I would consider using a slightly smaller indentation, though. Please also note that the figure is quite low resolution in the main text, if you were not already aware.

Figure 1 (as well as Figure 2) have been updated to a higher resolution.

– Figure 3: This forest plot is very helpful. I would encourage the authors to include more forest plots for other meaningful effects in the paper, as they will be of much greater relevance to the typical reader than the multiple, large funnel plots, which are important but could just as well be placed in the supplement and referenced from the main text.

Funnel plots have been moved to Supplementary files. Forest plots for all significant outcome domain associations have been added.

Tables– I am curious why Table 1 and 2 are presented as separate tables. It is a very long section in this article, and structuring it as clear as possible will be very helpful.

They have been combined (both Tables 1 and 2, and 3 and 4).

– Table 5: Please bold the checks to make them easier to see. Currently, the Xs stand out much more than the checks. Also, I would consider using a different color scheme to aid colorblind readers, as well as including the direction of effect change (i.e. + or -) next to the checks with the reference categories indicated in the footnote.

We’re grateful for the reminder of the need to use a color deficiency friendly palette. Moderators and reference categories have been specified for all analyses. We use + and – signs to indicate direction of significant effects, and as before, crosses for nonsignificant effects. Please note that it is hard to make significant associations ‘pop out’ more without making them bigger, which leads to formatting problems in the table.

– Is there a reason why the authors did not discuss the moderation effects from Table 5 in the Discussion section?

We did not include non-significant moderators (other than those we explicitly planned to focus on) for brevity. We now explicitly mention all significant moderators in text for outcome domains, and for brevity, refer readers to the Supplementary files for outcome measure moderations.

[Editors’ note: further revisions were suggested prior to acceptance, as described below.]

Senior Editor comments:1. This is a passage from essential revision #1 from the previous decision letter:"This would shift in the framing of the paper to be more of an exploration of the different patterns of data reported in this field, with more skepticism about reporting bias, rather than mainly a primary hypothesis test. The authors here have a terrific opportunity in this compiled dataset to examine if these hypotheses are actually testable (or to what extent) using available data. For example, do the observations between X and X trait hold true when excluding studies presenting this as their main hypothesis? In this way, their paper becomes more than just another manuscript of many that attempts to test extant hypotheses, and more of a critique and technical examination of the state of available data – which would be ultimately more useful across a range of disciplines."In my view, two of the most essential parts of this point was not accomplished in the revision. One, for a shift to a different type of paper that one necessarily aiming to draw biological conclusions. Two, for skepticism related to reporting bias (i.e. the tendency for negative results to not be published, or even for the effects of inadvertent or worse p-hacking in the published literature, coming into the meta-analyses).

We thank the editor for the greater clarity regarding their thinking on this issue provided through our email exchange and we append the full email history at the end of this document for reference. We believe that the conclusion of this exchange was that a focus on concerns around publication bias “could ultimately be the leading reason for this work to be featured in a major general life sciences journal, rather than a more field-specific journal, and to become a maximally impactful paper.”

On re-reading the paper we can see the advantages in more clearly spelling out the role of meta-analysis in general in this area, and have added a section starting at line 147 specifically on this (incorporating the previous meta-analyses on this specific topic that were already mentioned). This places our paper within that broader history of using meta-analysis to more clearly test evolutionary hypotheses with regard to humans and sexual selection.

Regarding the editor’s concern that meta-analyses addressing specific research questions are not suitable in *eLife*, we note that *eLife* has published prior examples of this kind of analysis. For instance, Baumelle et al., (2020, https://elifesciences.org/articles/55659) who tested specific hypotheses around biodiversity, and Mancini et al., (2020 https://elifesciences.org/articles/61523) who – like us – investigated which measures within a particular domain were and were not associated with a particular outcome (within neurohistology in their case). This demonstrates that theoretically driven meta-analyses of specific issues have a place within *eLife* alongside those that focus on methodological issues. We would also argue that a large-scale meta-analysis attempting to understand the level of selection for physical dimorphism in humans is of sufficient interest to a wide range of fields (including Psychology, Anthropology, and Biology) to justify its place within a general journal such as *eLife*.

While some meta-analysis in *eLife* have indeed focused on publication biases, such focus was also justified by the results. As we noted in our email exchange, our results do not support an exclusive focus on this angle and, as we further noted in response to the editor’s query and as we have now included in the discussion (lines 551-555), the main results do hold when only unpublished datasets were analysed. Publication bias is not, therefore, a primary concern with this research issue.

We hope that this provides both reassurance alongside our additional engagement with the goals of meta-analysis in our introduction, including concerns that publication bias may hinder interpretation, and can satisfy on this point.

2. Related to FDR q-values. Thank you for providing these in the revision. But how was this analysis performed? It is surprising that the q-values are as low as they are being reported, considering the large number of tests performed. I think that the most conservative approach should be used here (all p-values from all of the tests in the paper), given the implications and the large number of tests performed.

We believe this issue has been resolved through our email exchange, specifically:-

Our previous response by email: We did indeed run our correction for multiple comparisons in the suggested way: using all *p*-values from all of the tests in the paper (266 *p*-values in total). We include here output from the q value computation; as is evident from this output, where q values are low this is due to *p*-values being very small to begin with.

Editor’s further response: For the FDR analyses, I apologize for potentially missing the methodological statements in the text that you did indeed include all p-values into a single FDR calculation. That's great that you did do it this way.

We have also added a line specifying that this was how the q-value computation was performed (lines 291-292). Supplementary Figure 7 is also included in this resubmission; we apologise that it was erroneously left out from our previous resubmission.